# A convolutional neural network for the prediction and forward design of ribozyme-based gene-control elements

**Calvin M Schmidt[1], Christina D Smolke[1,2]\***

[1]Department of Bioengineering, Stanford University, Stanford, United States; [2]Chan Zuckerberg Biohub, San Francisco, United States

**Abstract** Ribozyme switches are a class of RNA-encoded genetic switch that support conditional regulation of gene expression across diverse organisms. An improved elucidation of the relationships between sequence, structure, and activity can improve our capacity for de novo rational design of ribozyme switches. Here, we generated data on the activity of hundreds of thousands of ribozyme sequences. Using automated structural analysis and machine learning, we leveraged these large data sets to develop predictive models that estimate the in vivo gene-regulatory activity of a ribozyme sequence. These models supported the de novo design of ribozyme libraries with low mean basal gene-regulatory activities and new ribozyme switches that exhibit changes in gene-regulatory activity in the presence of a target ligand, producing functional switches for four out of five aptamers. Our work examines how biases in the model and the data set that affect prediction accuracy can arise and demonstrates that machine learning can be applied to RNA sequences to predict gene-regulatory activity, providing the basis for design tools for functional RNAs.

## Introduction

The genetic engineering of novel biological systems has the ability to produce solutions to a wide array of global challenges (**Khalil and Collins, 2010**). Researchers have used genetically modified organisms as living medicines (**Anderson et al., 2006**), factories for specialty chemicals (**Steen et al., 2010**), and as tools for understanding biology (**Ferrari et al., 2007**). There are enormous potential economic and societal benefits that are dependent on our ability to design biological systems that function as intended, especially those that can make complex decisions necessary to function across the diverse conditions these systems may encounter by changing their gene expression profiles in response to environmental signals (**Mehrotra, 2016**).

Genetic switches comprise functional RNAs or proteins that change the expression of a target gene(s) conditional to the concentration of a specific molecule, or cognate ligand, to the switch (**Alberts et al., 2010**). Genetic switches are an important class of biological tools that allow cells to sense and respond to changes in their environment. In natural systems, they enable organisms to survive stressful conditions (**Oppenheim et al., 2005**), and in the hands of genetic engineers, genetic switches have enabled a diverse set of useful applications, from evaluating thousands of biological designs in a single experiment (**Michener and Smolke, 2012**) to creating safer therapies (**Ren et al., 2018**).

RNA switches are short, noncoding RNA sequences that act as genetic switches (**Schmidt and Smolke, 2019**). They work through a variety of mechanisms, including altering transcription termination (**Fowler et al., 2008**), transcript stability (**Ausländer et al., 2010**), and translation initiation (**Klauser et al., 2012**). Ribozyme switches, also known as aptazymes, are a special class of RNA switches that couple the binding of the ligand to an RNA aptamer to a change in the self-cleavage activity of a ribozyme (**Win and Smolke, 2007**). The sTRSV hammerhead ribozyme, one particular

**\*For correspondence:** csmolke@stanford.edu

**Competing interest:** The authors declare that no competing interests exist.

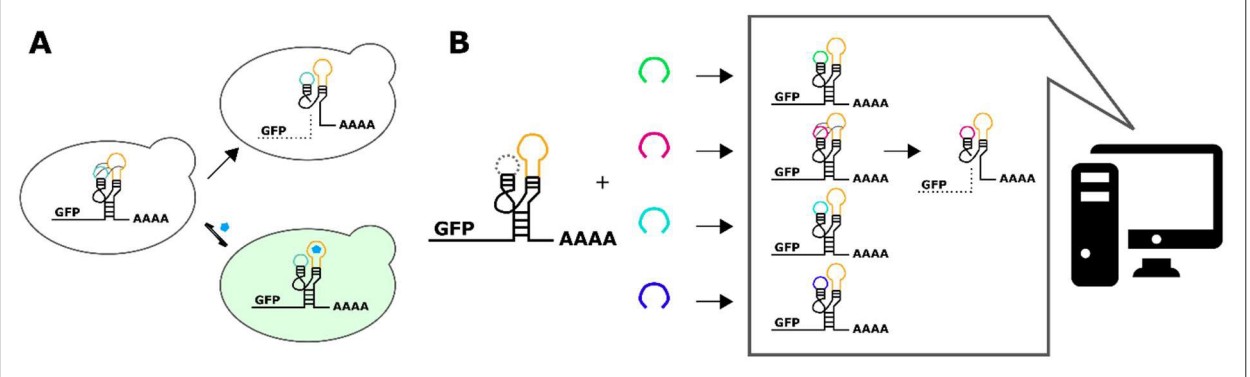

**Figure 1.** A computational workflow for the design of tertiary-interaction ribozyme switches. (**A**) Schematic of the mechanism by which tertiary-interaction ribozyme switches regulate gene expression. In the absence of ligand (blue pentagon), interactions between nucleotides in the variable loop (green) and the aptamer (yellow) lead to cleavage, separating the gene of interest from the poly-A tail, resulting in rapid degradation of the target transcript and a decrease in gene expression. Ligand binding to the aptamer disrupts the loop-loop interactions, thereby disrupting cleavage and increasing transcript stability and gene expression. (**B**) Schematic of the computational design process for a tertiary-interaction ribozyme switch. The selected aptamer is grafted onto one of the two ribozyme loops, and a library of variable loops (varied in both length and sequence) is grafted onto the opposite loop. A computational model then analyzes each member of the library in silico for predicted gene-regulatory activity; sequences predicted to result in low gene-regulatory activity in the absence of ligand can be synthesized and subsequently characterized in vivo.

ribozyme, is a short sequence of RNA that self-cleaves when it adopts a specific tertiary conformation based on electrostatic interaction between nucleotides in two of its loops. The ribozyme switch is integrated into a transcript such that changes in cleavage activity result in changes to transcript stability and thus gene expression (*Figure 1A*). In tertiary-interaction ribozyme switches, one of the two loops is fixed as an RNA aptamer, and the sequence of the opposing loop (the variable loop) is selected so that the nucleotide interactions necessary for cleavage at physiological conditions occur in the basal state (or absence of ligand) (*Townshend et al., 2015*). In optimizing a tertiary-interaction ribozyme switch for gene expression control, one typically selects for a switch with very high cleavage activity in the absence of ligand, corresponding to low basal gene expression, expecting that the ligand binding to the aptamer will block the electrostatic interactions that are necessary for cleavage. This approach for switch selection may allow for the forward design of tertiary-interaction ribozyme switches by using computational models that predict loop sequences that pair with a given aptamer sequence to result in high levels of cleavage in the absence of ligand (*Figure 1B*).

Several computational tools for RNA switch design have been described that use various strategies to generate RNA-based genetic switches from a starting aptamer. In one example, a software tool was developed that computationally iterates through thousands of designs to identify a subset predicted to form secondary structures that may result in ligand-dependent transcription termination (*Domin et al., 2017*). 17 switches were generated for three aptamers, and 12 of those (71%) exhibited significant changes in expression in the presence of the cognate ligand with activation ratios, defined as the ratio in gene-regulatory activities in the presence and absence of ligand, up to 4. In another example, a biophysical model of ligand binding and RNA secondary structure conformation was developed to generate switches that occlude the ribosome binding site when the aptamer is not bound to the ligand (*Espah Borujeni et al., 2016*). The tool's predictions of activation ratio for 59 switch sequences fell within twofold of the measured activation ratio 63% of the time, and select switches reached activation ratios of over 100.

While the above examples describe computational tools that support the design of RNA switches that function in bacterial systems such as *Escherichia coli*, there are a number of switch platforms that have been demonstrated in eukaryotic host organisms such as yeast or mammalian cells, with corresponding applications pursued. One computational method for switch design in mammalian systems leveraged a biophysical model of ligand binding and RNA secondary structure conformation to design inducible RNAi systems in HEK cells (*Beisel et al., 2008*). The switches designed with this tool exhibited activation ratios of up to 5. However, none of the computational design tools described to date are based on gene-regulatory platforms that function across a broad range of organisms (*Xiang et al., 2019*). We set out to fill this void by creating genetic switch design software

for a ribozyme switch platform that has been demonstrated in both bacterial and eukaryotic cells and across diverse applications (*Ausländer et al., 2010*; *Furukawa et al., 2014*; *Liang et al., 2012*). While other software tools exist for designing RNA switches, ours is the first that works using the ribozyme platform. This is important not only because this platform has been demonstrated across different cell systems, but also because, to advance our ability to design biological systems, having orthogonal systems of gene regulation allows for larger constructs to be built without genetic overlap, thus avoiding recombination-based dropout. In addition, the other design tools have not used machine learning, but rather biophysical models and rules for their design process. One of the benefits of a machine learning-based model is that it will only improve as the quantity and quality of the training data is improved.

Here, we show that machine learning can predict the activity of a hammerhead ribozyme when given the sequence of that ribozyme. We developed a training data set for the model by constructing a library of 150,000 unique ribozymes and used a high-throughput FACS-Seq screening method to measure the gene-regulatory activities of individual library sequences. We demonstrated that a convolutional neural network (CNN) can learn how the sequence of a ribozyme determines its gene-regulatory activity with high accuracy for ribozymes with small loops ($R^2 = 0.91$) and medium accuracy for ribozymes with an aptamer as one of the two loops ($R^2 = 0.28$). We used the resulting model to design a set of novel ribozyme switches using five different aptamers that alter gene expression upon a change in cognate ligand concentration in yeast cells. The model produced ribozyme switches with basal gene-regulatory activities lower than 30% of that from a nonfunctional ribozyme control (sufficient to enable activation ratios of at least 3) for four of the five aptamers and ribozyme switches that exhibit significant changes in gene-regulatory activity in response to the cognate ligand for all aptamers tested, proving that this model can be useful as a forward design tool for genetic switches. We have packaged the model into a design tool that accepts an aptamer sequence and returns a set of ribozyme switch candidate sequences for testing. The design tool can be used by the broader research community to design genetic switches tailored to ligands of interest for which aptamer sequences are available, advancing the field of synthetic biology by enabling the computational design of dynamic biological systems.

## Results

### High-throughput measurement of ribozyme gene-regulatory activity generates a diverse training data set

In order to generate a model that can predict gene-regulatory activity from ribozyme sequence, we first generated a data set of diverse ribozyme sequences and their gene-regulatory activities to train the model. We wanted the designed training set library to contain ribozymes with diverse sequences and structures so that the model would be able to learn which nucleotide interactions between the loop regions can support cleavage in physiological environments. In addition, because the model would be used to design ribozymes that exhibit low gene-regulatory activities in the absence of ligand, it was important that the training set contain thousands of ribozymes with low gene-regulatory activities, which are normally rare in ribozyme libraries with randomized sequences.

We designed two ribozyme libraries to capture the desired diversity and cleavage activities. The first library was designed by randomizing the loop regions (loops I and II) with nucleotide sequences of lengths 5–10 on the base of sTRSV hammerhead ribozyme scaffold (*Figure 2A*). Libraries were cloned into the 3' untranslated region (UTR) of a GFP expression cassette encoded on a low-copy plasmid, such that ribozymes with high cleavage activities result in cells expressing low GFP levels. A constitutive mCherry expression cassette was included on the ribozyme library screening plasmid and used to normalize for intrinsic gene expression noise due to factors like cell cycle state and plasmid copy number (*Liang et al., 2012*).

The gene-regulatory activities of individual ribozyme sequences in the libraries were measured via FACS-Seq(*15*) (*Figure 2B*). FACS-Seq uses fluorescence-activated cell sorting (FACS) to partition cells harboring libraries of regulatory elements into different activity bins based on the cell's ratio of GFP to mCherry levels and next-generation sequencing (NGS) to determine the number of times a ribozyme sequence occurs in a particular bin. For a given sequence, the distribution of counts across the bins is used to generate a mean measured value of gene-regulatory activity associated with that

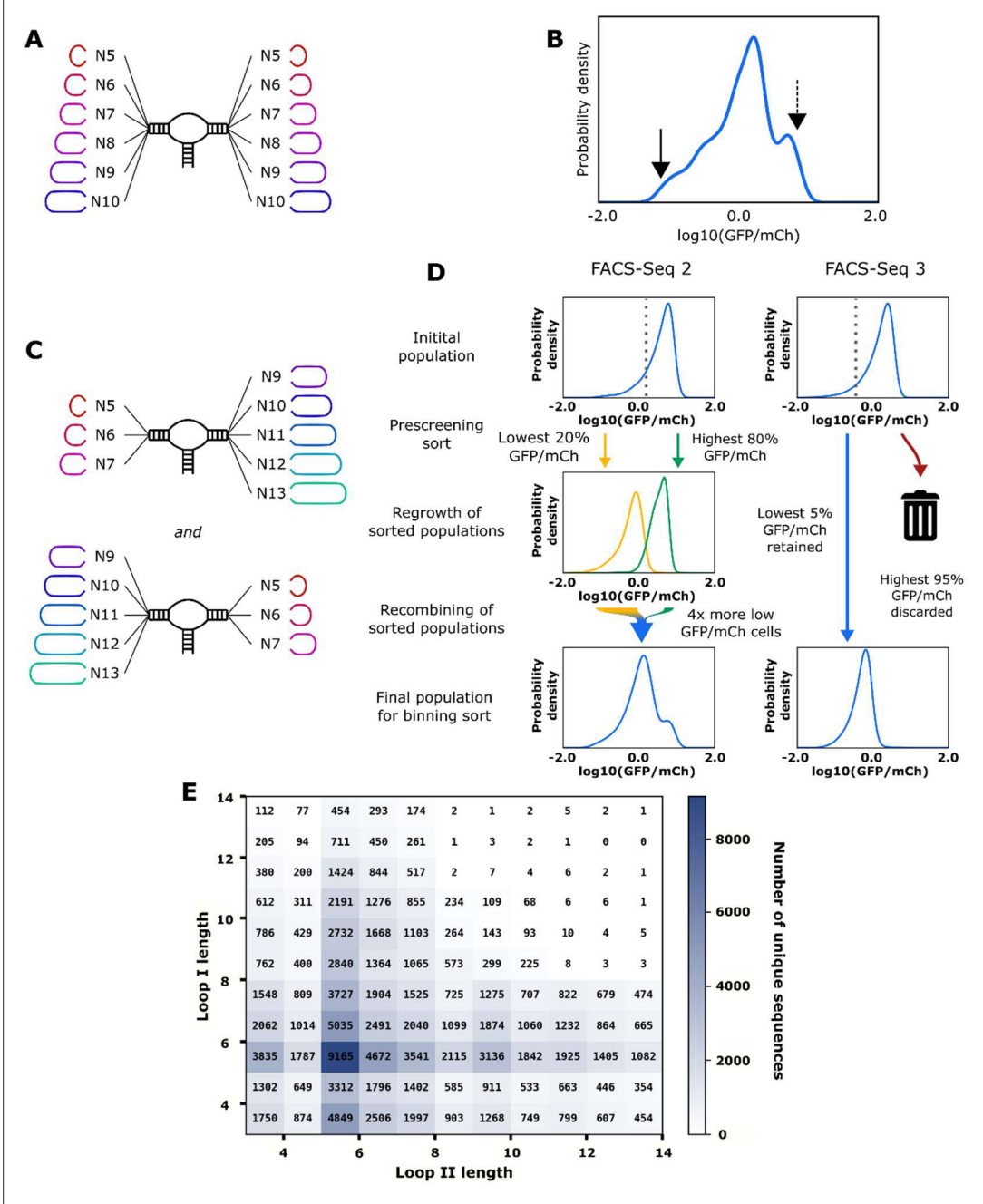

**Figure 2.** Design and high-throughput characterization of ribozyme libraries with random variable loop sequences. (**A**) Schematic of the design of ribozyme libraries for the model training data set FACS-Seq 1. (**B**) The distribution of gene-regulatory activities for ribozyme libraries analyzed in FACS-Seq 1. A mean gene-regulatory activity (log10(GFP/mCherry)) value for each sequence is generated using the distribution of that sequence among the different FACS-Seq activity bins (see Materials and methods). The solid arrow indicates the mean gene-regulatory activity of the native ribozyme (sTRSV) control, and the dashed line indicates the mean gene-regulatory activity of the inactive ribozyme (sTRSVc) control. FACS-Seq 1 consists of approximately 75,000 unique ribozyme sequences, for which at least 20 reads for each sequence were recorded. (**C**) Schematic of the design of ribozyme libraries for the model training data sets FACS-Seq 2 and FACS-Seq 3. (**D**) The distributions of gene-regulatory activities for prescreened ribozyme libraries analyzed in FACS-Seq 2 and FACS-Seq 3. The initial FACS-Seq 2 library was sorted into a low- and high-expressing population by setting the bin barrier at log10(GFP/mCh) = 0.28, the 20th percentile value. Each population was passaged for 2 days, then recombined for the binning sort. When recombining, four times as many cells from the low-expressing population as the high-expressing population were

*Figure 2 continued on next page*

*Figure 2 continued*

included in the final population, enriching for low basal gene-regulatory activity ribozymes. FACS-Seq 2 consists of approximately 61,000 unique ribozyme sequences. The initial FACS-Seq 3 library was sorted into a low- and high-expressing population by setting the bin barrier at log10(GFP/mCh) = –0.22, the 5th percentile value, but only the low-expressing population was retained for binning. FACS-Seq 3 consists of approximately 9000 unique ribozyme sequences. (**E**) The distribution of sequences analyzed in the FACS-Seq experiments for acquiring the training data set as a function of loop size.

The online version of this article includes the following figure supplement(s) for figure 2:

**Figure supplement 1.** Replicate data for FACS-Seq runs 1–4.

---

ribozyme. The first designed ribozyme library was transformed into *Saccharomyces cerevisiae* cells and FACS-Seq was used to obtain gene-regulatory activity values for approximately 75,000 ribozyme sequences with at least 20 reads each; the resulting data set is designated as FACS-Seq 1 (*Figure 2—figure supplement 1A*). 20,000 of these ribozymes have at least one loop with a length of ≥8, which is the size of the smallest aptamer loops. 48% of the sequences exhibit activity that is greater than half of that exhibited by a nonfunctional ribozyme, and only 7% (1432 sequences) displayed activity near that of the native sTRSV hammerhead ribozyme. The activity data indicates that most of the possible nucleotide sequence combinations for loops I and II do not result in functional nucleotide interactions that support high cleavage activity at physiological conditions.

The second library was designed and characterized to ensure a larger fraction of low gene-regulatory activity ribozyme sequences. The library was designed with one of the loops (loop I or II) harboring a randomized sequence of larger length (N9–N13) and the other loop (loop II or I, respectively) harboring a shorter randomized sequence (N5–N7). The design of the larger loop was meant to capture contexts in which an aptamer is placed onto either loop I or II, such as in tertiary ribozyme switch designs, and to allow the model to extend to the design of those more complex architectures (*Figure 2C*).

The second library was characterized through two modified FACS-Seq experiments designed to increase the representation of low gene-regulatory activity ribozymes, producing the data sets FACS-Seq 2 and FACS-Seq 3. In FACS-Seq 2, cells harboring the ribozyme library were pre-sorted by separating out the cells with the highest 80% expression levels from those with the lowest 20%. The two populations were cultured separately for several days and then combined in a ratio such that the low-expressing cells comprised 80% of the final population (*Figure 2D*), which was subsequently analyzed by FACS-Seq. This recombined population, FACS-Seq 2, still contained ribozymes that spanned a broad range of expression (*Figure 2—figure supplement 1B*). Of the 31,000 sequences that had a loop length of ≥8, even with the prescreening, 24% still exhibited high gene-regulatory activities (greater than half of the activity exhibited by a nonfunctional ribozyme). However, although the loops of this library were of much larger size, which generally leads to high gene-regulatory activity (*Townshend et al., 2015*), another 1,424 low -activity sequences were isolated, thus doubling the number of ribozymes in our training set with gene-regulatory activity close to that of the native ribozyme. To obtain additional training data on low gene-regulatory activity ribozymes, we subjected the second ribozyme library to a more stringent pre-sort prior to the FACS-Seq analysis. Specifically, we separated out the cells with the lowest 5% expression levels and ran this cell population through the FACS-Seq protocol, creating another training set designated FACS-Seq 3 (*Figure 2—figure supplement 1C*). Due to the stringent pre-sorting process, the final library was enriched in low gene-regulatory activity ribozymes, adding another 1,158 (of the 5,000 sequences in FACS-Seq 3 that had a loop length of ≥8) unique low gene-regulatory activity ribozymes with large loops to the training set.

Through the design and testing of multiple ribozyme libraries, we generated individual measurements of gene-regulatory activities for roughly 150,000 unique ribozyme sequences (10,000 of which exhibit gene-regulatory activities close to that of the native ribozyme), achieving a data set with a broad range of activities. The libraries contain diverse sets of loop structures, with many combinations of loop sizes containing hundreds to thousands of unique sequences (*Figure 2E*). Loops ranged in size from 3 nucleotides (e.g., an N5 loop where the first and last nucleotides could base pair) to 13 nucleotides. Of the larger loops (length ≥10 nucleotides), between ⅛ and ½ of the loops in the training set contain a stem branching off of the loop, providing significant structural diversity to our training set. The library also contains a diversity of stem lengths, with the first stem (normally 6 base pairs long)

extending up to 11 base pairs and the second stem (normally 4 base pairs) extending up to 8 base pairs. Approximately 60,000 of the ribozymes in the characterized library have a loop with length >8, and 4,000 of these large-loop sequences displayed near-native gene-regulatory activity. The thorough data sets generated from this work can be used to train a robust model for predicting ribozyme activity from sequences spanning diverse loop sizes and structures.

## A convolutional neural network accurately and robustly predicts gene-regulatory activity from ribozyme sequence

Before assessing different machine learning models for ribozyme design, we developed a method by which the features of a given ribozyme (i.e., sequence, structure) are encoded for input into the model. Given the mechanisms underlying ribozyme cleavage, it is important for the encoding to retain as much information as possible about the physical system and the relationship between the features. For example, when performing image analysis, the data is kept in a two-dimensional array (with extra dimensions added to represent color) that retains the physical relationship between different areas of the image.

Our approach to encoding the ribozyme features focuses on retaining the features' physical relation to each other. The method starts by using the RNA secondary structure prediction program RNAstructure (*Reuter and Mathews, 2010*) to determine the structure (and therefore the sequence) of loops I and II of the ribozyme (*Figure 3A*). While the sequence of the ribozyme is known, which nucleotides comprise a loop is not known until the structure is determined due to the fact that different loop sequences can produce different loop structures. One such example is if the first two nucleotides in the loop after the stem can form a Watson–Crick pair, which would extend the stem further and therefore make the following two nucleotides the actual first ones in the loop. Another example is ribozymes with longer loop sequences that can form stems extending off of the loop, which functionally removes some of the nucleotides from the loop. Once the loop sequence is determined, turning stems I and II and loops I and II of the ribozyme are turned into an image with two spatial dimensions, each nucleotide occupying one 'pixel' and another dimension that represents the base and whether or not it is paired (*Figure 3B*). To simulate the stem-loop I and stem-loop II interactions, the two-dimensional images that represent stem-loops I and II are paired with each other, creating a three-dimensional (3D) image that represents the stem-loops. The described encoding method preserves the spatial orientation of the nucleotides in stem-loops I and II and includes all nucleotides in the stem-loops for analysis. One potential drawback of the encoding method is that it makes some assumptions regarding the spatial arrangement of the nucleotides, specifically that the stem-loop nucleotides occupy a relatively thin plane, which extends in one direction as the loop grows larger. Evidence from crystal structures for the native hammerhead ribozyme supports this assumption of the nucleotides being arranged in two thin opposing planes (*Chi et al., 2008*).

The described ribozyme encoding method is optimally paired with a CNN. The algorithm begins by analyzing small groups of features at a time before combining them to make a final prediction. In our model, we applied a 3D convolution that was two nucleotides long along each edge, creating a window that captures each group of eight adjacent nucleotides (*Figure 3C*). The window scans along the length of the array, moving one nucleotide at a time, creating a set of one-dimensional arrays. The one-dimensional arrays are converted to a single prediction through a set of fully connected layers and dropout layers. The resulting model is sufficiently complex to learn which sequences affect cleavage, while being simple enough that it does not overfit to the training data with a low number of training epochs (*Figure 3—figure supplement 1*). The main disadvantage of a neural network model is that it is not transparent in how it makes its predictions, such that it is difficult to obtain mechanistic insight into ribozyme cleavage from the model. However, model interpretability is a secondary concern to accuracy and robustness to structural differences as the primary function of the model is to design new ribozyme switches given a known aptamer sequence.

As it is generally preferred to use as simple a model as possible for a given objective, we compared the performance of the described model with a version of a simple linear regression model known as a lasso model (*Figure 3—figure supplement 2*). We devised a series of tests to determine which model would be most useful for the predictive design of ribozyme switches. The model's objective is to make accurate predictions of basal gene-regulatory activities for ribozymes that have an aptamer replacing either loop I or II.

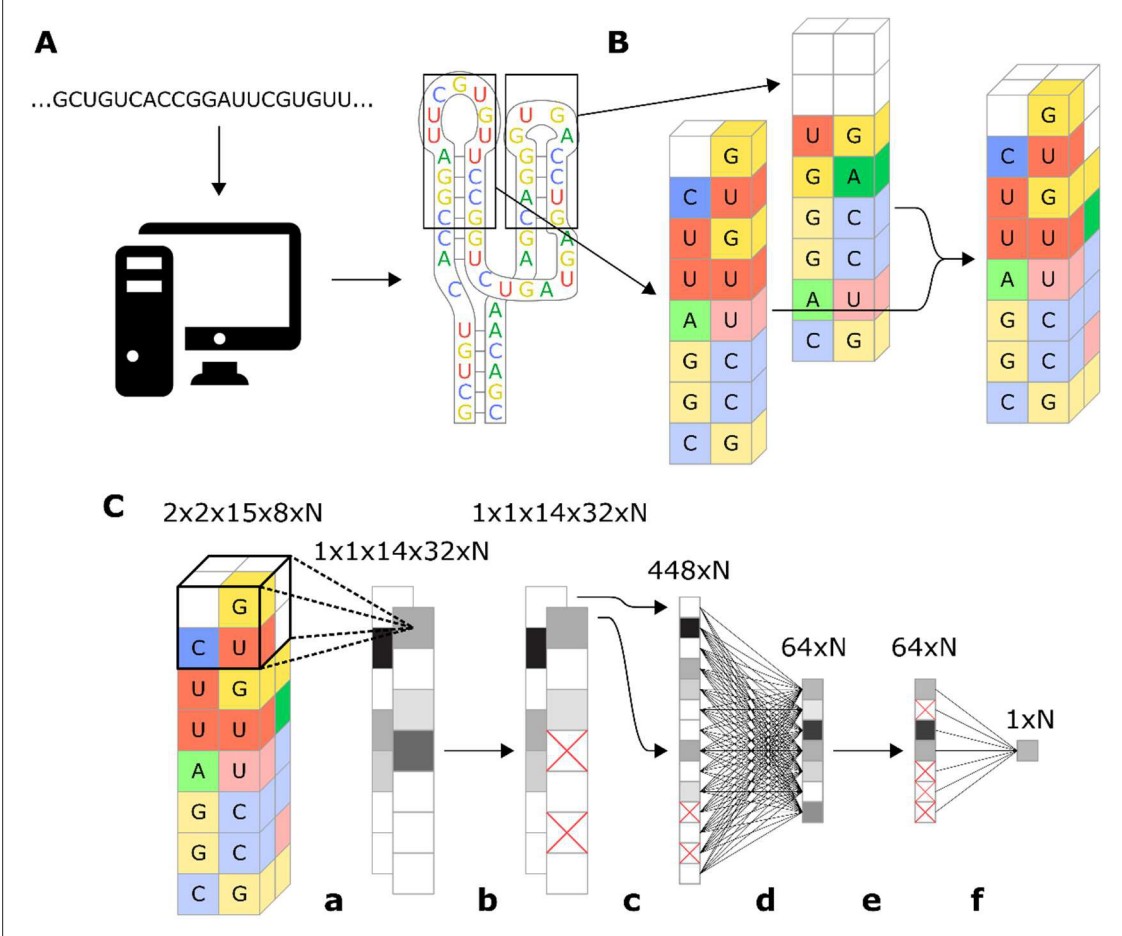

**Figure 3.** Workflow for encoding of a ribozyme sequence into numerical features and the architecture of the neural network that converts those features into a predicted gene-regulatory activity. (**A**) The RNA sequence is computationally folded to obtain a predicted secondary structure. (**B**) The predicted secondary structure is converted into a three-dimensional set of features, where each feature is a binary recording across eight channels that represent the four nucleotides and being either paired or unpaired. For example, in the figure the first nucleotide of the loop is a U, so the feature that represents that position is given a value of 1 in the unpaired U channel, and a 0 in the rest of the channels. Positions that have no nucleotide occupying them are given values of 0 for all channels. (**C**) Schematic representation of the model architecture of the convolutional neural network that predicts gene-regulatory activity. The numbers above each layer demonstrate the dimensions of the array of N sequences at each step of the network. The layer order is: (**a**) 2 × 2 × 2 convolution with step size of 1, (**b**) 25% dropout, (**c**) flatten, (**d**) fully connected with rectilinear activation, (**e**) 50% dropout, and (**f**) fully connected to reduce to one decimal value.

The online version of this article includes the following figure supplement(s) for figure 3:

**Figure supplement 1.** Training and test loss curves for the neural network.

**Figure supplement 2.** Workflow for encoding of a ribozyme sequence into numerical features with a linear regression algorithm.

We first compared the two models on accuracy of predicting gene-regulatory activity from ribozyme sequence. The data set of roughly 150,000 ribozyme sequences with measured gene-regulatory activities was randomly separated into a training subset (75% of the sequences) and a test subset (25% of the sequences). The neural network model and lasso model were trained on the training subset using backpropagation to minimize mean-squared error on the predicted log10(GFP/mCherry) value when compared to the measured log10(GFP/mCherry). The trained models predicted the gene-regulatory activities of ribozymes in the test subset. A model's accuracy was evaluated by generating an $R^2$ value when analyzing the measured and predicted basal gene-regulatory activity values. We first tested the model's ability to predict the simplest ribozymes we had measured data on in order to establish a baseline of predictive power. These were ribozymes with four random nucleotides on each loop (*Figure 4A*). Both models were able to predict the activities in the simulated test subset (25% of the constructed training set) with high accuracy; however, the neural network ($R^2$ = 0.91) was more

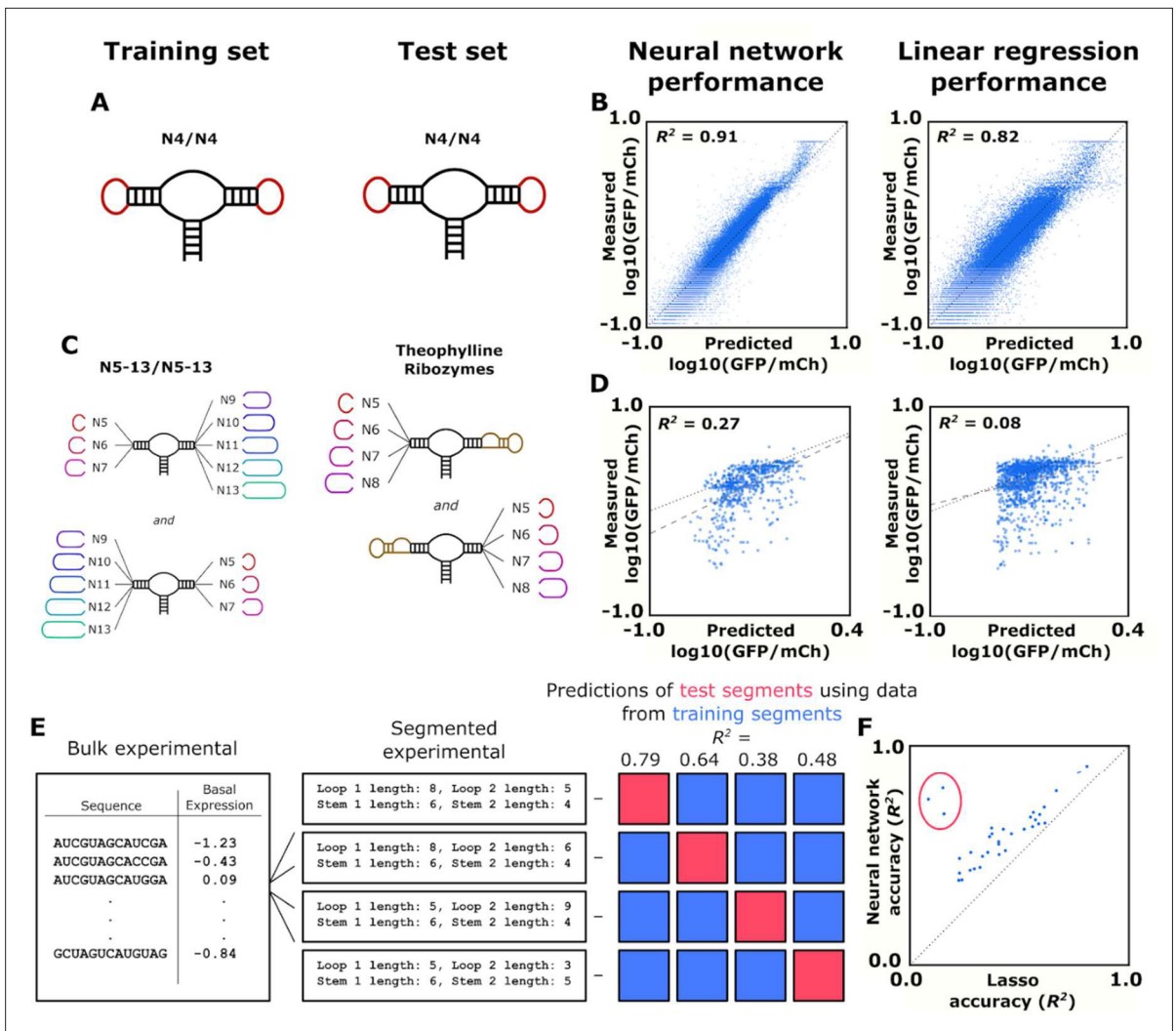

**Figure 4.** Training and validation of models for predicting ribozyme gene-regulatory activity. (**A**) Schematic of the training and test sets for simple ribozymes, wherein both sets comprise ribozymes with four random nucleotides on each loop. No ribozyme sequences are shared between the two sets. (**B**) Comparison of the neural network and linear regression algorithms on their ability to predict the basal expression of simple ribozymes. Each point on a plot represents the predicted and measured basal gene-regulatory activity for a single ribozyme sequence. The points for accurately predicted ribozymes fall close to the dotted line that represents when the predicted and measured basal gene-regulatory values are equal. (**C**) Schematic of the training and test sets for ribozymes with a theophylline aptamer in place of one of the loop regions. The models were trained on ribozymes with four random nucleotides on each loop. The trained models were used to predict the basal gene-regulatory activity of ribozymes with a theophylline aptamer in place of one of the two loop regions. (**D**) Comparison of the neural network and linear regression algorithms on their ability to predict the basal expression of ribozymes with a theophylline aptamer whose activity was measured through FACS-Seq. Each point on a plot represents the predicted and measured basal gene-regulatory activity for a single ribozyme sequence. The black dotted line represents the where predicted activity equal the measured activity for a given ribozyme, while the gray dashed line represents the line of best fit to the predicted data. (**E**) Workflow for the evaluation of an algorithm's accuracy when predicting the basal gene-regulatory value of ribozymes whose structure differs from the structures in the training set. The sequences in the experimental data set are first grouped into different segments based on the size of the loops and the stems in the computationally predicted secondary structure such that each ribozyme in a segment shares the same structure. The gene-regulatory activity levels of the sequences in one segment (the test segment) are predicted using a model trained on the sequences in other segments (grouped into a training segment). This prediction is compared to the experimental data to determine the accuracy, as measured by the $R^2$ value, of predicting that segment based on segments that are structurally different. This process is repeated using each segment as the test segment. (**F**) Comparison of the ability of two algorithms, a lasso linear regression model (Lasso) and a three-dimensional convolutional neural network, to predict the basal gene-regulatory activities of different test segments. Each segment is represented by a single point, with its positioning determined by the $R^2$ value of a model's prediction for sequences in that segment. The dotted line represents the point where the two models would have equal accuracy. The points encompassed by the red circle represent the segments that do not have the same stem length as the native ribozyme.

The online version of this article includes the following figure supplement(s) for figure 4:

*Figure 4 continued on next page*

*Figure 4 continued*

**Figure supplement 1.** Random forest and SVM prediction of ribozyme gene-regulatory activity.

**Figure supplement 2.** Model accuracy as a function of the size of the training set.

**Figure supplement 3.** Model accuracy as a function of the similarity of structures in the test set to those in the training set.

accurate than the lasso algorithm ($R^2$ = 0.82). (*Figure 4B*). In addition to these predictive methods, we also tested a support vector machine ($R^2$ = 0.45) and a random forest ($R^2$ = 0.92) algorithm on the same N4N4 data set (*Figure 4—figure supplement 1A*). The random forest also scored well in its prediction of the gene-regulatory activity of simple ribozymes. Although they had a better $R^2$ value overall, ribozymes that were predicted to have low gene-regulatory activity by the neural network model were equally likely to have low measured gene-regulatory activity than those predicted by the lasso model (mean log10(GFP/mCherry) value of –1.41 to –1.40 for the 10 ribozymes with the lowest predict gene-regulatory activity).

We next evaluated the model's ability to predict gene-regulatory activities of ribozymes with more complex structures. An earlier study performed a high-throughput screen on ribozymes with an aptamer integrated into one of the two loops of the sTRSV hammerhead ribozyme (*Townshend et al., 2015*). We used that published data set as a test set for evaluating the model's predictive ability on more complex ribozyme structures. In order to simulate the design of a novel ribozyme switch, we used ribozymes with a theophylline aptamer as one loop for a test set for predictions of gene-regulatory activity in the absence of ligand (*Figure 4C*). We used the trained neural network model and the lasso model to predict the basal gene-regulatory activity of each sequence in the theophylline ribozyme switch data set and compared the predicted activities with measured values as described above (*Figure 4D*). Both models were less accurate in predicting the activities of more complex ribozyme sequences than when predicting the activities of N4N4 ribozymes; however, the neural network model generated an $R^2$ value that was much higher than the $R^2$ value produced by the lasso model (0.27 vs 0.08). As before, we also tested a support vector machine (SVM) ($R^2$ = 0.002) and a random forest ($R^2$ = 0.06) algorithm, which performed worse than either the neural network or lasso algorithms (*Figure 4—figure supplement 1B*). The neural network model was also more accurate in predicting which ribozymes would display low gene-regulatory activity. The 10 ribozymes with the lowest neural network-predicted activities had a mean log10(GFP/mCherry) value of –1.04, while the best lasso-predicted ribozymes had a mean log10(GFP/mCherry) value of –0.69. This example offers an approximation to how the models would perform when designing novel ribozyme switches, and the data indicate that the neural network model has more accurate predictive power.

In addition to the level of accuracy, it is also desirable for the model to be robust to differences in loop structures. The ribozymes comprising the training data set exhibit a variety of loop lengths, but aptamers can comprise structures that are more complex than a simple stem-loop. We therefore compared the two models in the accuracy of their activity predictions for ribozymes that have loop structures that differ from the ribozymes from which they have been trained on. To do this, we grouped the ribozymes from our training data set into different segments based on the size of the loops and length of the stems of each ribozyme such that each ribozyme in a segment shares the same structure (*Figure 4E*). We then took a group, held it out as a test set, and combined the remaining segments into a data set that was used to train the two models, which were then used to predict the gene-regulatory activities of the test set. This was repeated for each segment of ribozymes with a shared structure in our data set, generating a measure of accuracy ($R^2$) for each model. For each test segment of ribozymes with a shared structure, the neural network's predictions had an $R^2$ value an average of 0.18 greater than the predictions from the lasso model (*Figure 4F*). Furthermore, the neural network's $R^2$ values were higher than the lasso model for every single structural segment, and the segments where the stem length was different than the standard ribozyme stem had an even greater difference in $R^2$ (0.53).

Taken together, the results from the testing of the neural network and lasso model indicate that the former is the better choice from a forward design standpoint due to making more accurate predictions and being more robust to novel loop structures than the lasso model. Additional optimization was performed to determine the optimal size of the training set (*Figure 4—figure supplement 2*) and the effect of the structural difference between ribozymes in the training set and the test set on

accuracy (*Figure 4—figure supplement 3*). It is particularly important to note that the accuracy of the neural network model declines rapidly as the structure of ribozymes in the training set differs from the structure of the ribozymes whose gene-regulatory activity the model is predicting (*Figure 4—figure supplement 3C*). These properties will be important in developing a model to design ribozymes with aptamers as one of the two loops, as those sequences and specific structures are not necessarily present in our training data set. However, the diverse set of structures in the training set is to our advantage as it increases the chance that there are shared structures between the training set of ribozymes and the ribozymes with aptamers.

## The neural network model designs tertiary-interaction ribozyme switch libraries enriched for sequences with low gene-regulatory activity

We next examined the model's ability to perform the forward design of tertiary-interaction ribozyme switch libraries. Specifically, by using the model to screen in silico for sequences that have a greater likelihood of exhibiting low gene-regulatory activities in the absence of ligand, these sequences can comprise a smaller 'designed library' that can subsequently be efficiently screened in the presence and absence of the cognate ligand for high-performing switches (*Townshend et al., 2015*). A model that can determine which switches are more likely to exhibit low basal gene-regulatory activity can be used to design libraries enriched for sequences that are more likely to be switches, enabling a more effective screen. The model was tested with five small molecule aptamers, specifically sequences that bind folinic acid (*Chang, 2014*), theophylline (*Zimmermann et al., 1998*), tetracycline (*Berens et al., 2001*), chloramphenicol (*Burke et al., 1997*), and neomycin (*Weigand et al., 2008*). The aptamer-ligand pairs span a diverse set of chemical structures (*Figure 5A*), RNA sequences, and secondary structures (*Figure 5B*), enabling us to examine the ability of the model to design ribozyme switch libraries for diverse aptamer sequences.

The libraries for each of the five aptamers were designed similarly. The ligand-binding loop in the aptamer and any stems extending off that loop were attached directly to either stem I or II to create loop I or II. The loop opposite the aptamer loop was set as all possible sequences comprising N5–N8 to create a library of 174,080 ribozyme sequences per aptamer (*Figure 5C*). The model evaluated each sequence generated in this library in silico using the neural network model (trained on the generated training data set) to assign a predicted gene-regulatory activity (in the absence of ligand) value to each sequence (*Figure 5—figure supplement 1*). As the ribozymes used in the training set did not contain an aptamer structure on either loop, the model had to extrapolate its learnings to a new ribozyme sequence and structure. We ranked the examined sequences for each aptamer by their predicted gene-regulatory activity values and selected the 200 sequences with the lowest predicted values as our 'designed library'.

The activities of every member of the designed libraries for each aptamer were characterized in a single FACS-Seq experiment, FACS-Seq 4. The 200 sequences comprising each aptamer's designed library were combined into a 1,000 sequence library that was synthesized using a custom oligonucleotide pool. In addition to the designed library, we also constructed a random library using oligos encoding N5–N8 for the variable loops. The random library and the designed library were combined and analyzed through a FACS-Seq experiment. Through the FACS-Seq analysis, we obtained basal gene-regulatory activity values for 90% of the sequences from the designed libraries across all aptamers and nearly 20,000 sequences from all the aptamers in the random libraries (*Figure 2—figure supplement 1D*).

We compared the distribution of gene-regulatory activity values in the absence of ligand for sequences in the designed library with sequences in the random library to assess the model's ability to design ribozymes that exhibit low levels of basal gene-regulatory activity for each aptamer (*Figure 5D*). We evaluated the libraries using three principal metrics: (i) the mean gene-regulatory activity for sequences in each library (log10(GFP/mCherry)); (ii) the fraction of sequences with basal gene-regulatory activities in the lower third of the expected activity range (bounded by the gene-regulatory activity a fully functional and fully non-functional ribozyme, i.e., below log10(GFP/mCherry) = 0.0); and (iii) the statistical confidence measurement via a Wilcoxon Rank-Sum test, indicating if the designed and random libraries were drawn from different underlying distributions (*Table 1*). For all aptamers, sequences exhibiting low gene-regulatory activity in a random library are rare; no more than 3% of sequences in any random library exhibit basal gene-regulatory activity levels in the lower third of the activity range.

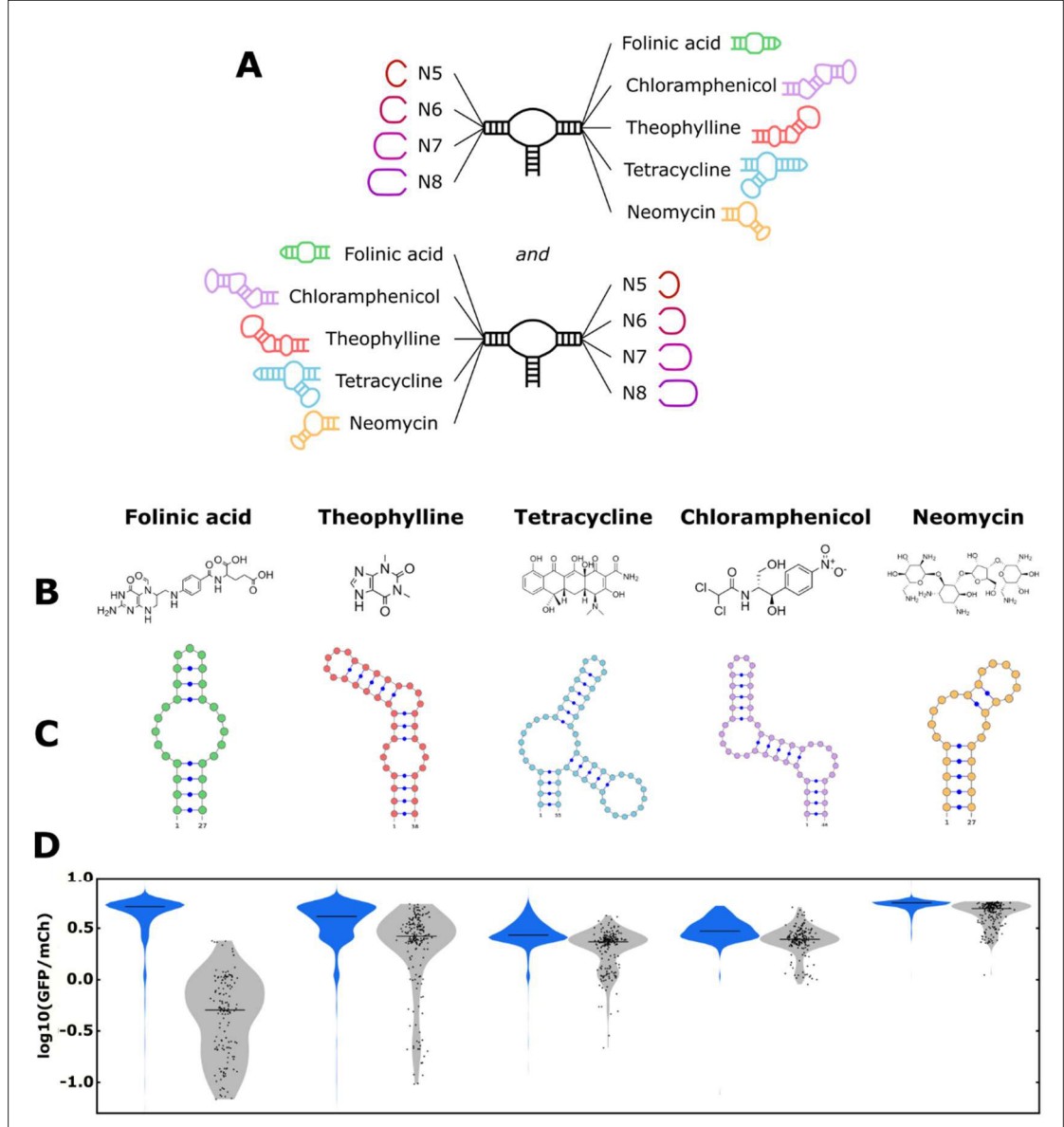

**Figure 5.** The convolutional neural network (CNN) model designs ribozyme libraries with aptamers grafted onto a loop region enriched in sequences exhibiting low gene-regulatory activities. (**A**) Schematic of the designs for the aptamer-ribozyme libraries. Each ribozyme has one variable loop, consisting of between 5 and 8 random nucleotides, and one aptamer loop (as indicated). (**B**) Chemical structures of the different ligands for which aptamers were tested in the model. (**C**) Schematic of the RNA secondary structures for the aptamers tested in the model. (**D**) Distribution of gene-regulatory activities across random and CNN-designed ribozyme libraries. The gene-regulatory activities of individual ribozyme sequences were measured via FACS-Seq. Blue violin plots: distribution of activities for the random libraries; gray violin plots: distribution of activities for the designed libraries. The individual activities of sequences in the designed libraries are shown as black points. Only sequences with more than 20 reads recorded were counted. For the random libraries, 5,264 folinic acid, 4,639 theophylline, 2,002 tetracycline, 911 chloramphenicol, and 1,417 neomycin sequences were counted. For the designed libraries, 154 folinic acid, 198 theophylline, 197 tetracycline, 174 chloramphenicol, and 196 neomycin sequences were counted.

The online version of this article includes the following figure supplement(s) for figure 5:

**Figure supplement 1.** Workflow for predicting ribozyme activity.

**Figure supplement 2.** Schematic detailing the process of adding an aptamer onto one loop of the ribozyme.

**Table 1.** Statistics on the gene-regulatory activities from the parent and designed libraries.
Library data is grouped based on aptamer, and the distributions are analyzed to determine the mean gene-regulatory activity value and fraction of sequences in the library that fell below a threshold gene-regulatory activity value (set as log10(GFP/mCh) = 0).

| | Folinic acid | Theophylline | Tetracycline | Chloramphenicol | Neomycin |
|---|---|---|---|---|---|
| Mean log10(GFP/mCh) value for parent library | 0.624 | 0.553 | 0.443 | 0.485 | 0.726 |
| Mean log10(GFP/mCh) value for designed library | −0.361 | 0.312 | 0.296 | 0.383 | 0.641 |
| Fraction below log10(GFP/mCh) = 0.0 for parent library | 0.027 | 0.024 | 0.012 | 0.001 | 0.001 |
| Fraction below log10(GFP/mCh) = 0.0 for designed library | 0.772 | 0.151 | 0.076 | 0.006 | 0.0 |
| Result of rank-sum test (p-value) | $1.2 \times 10^{-91}$ | $1.0 \times 10^{-34}$ | $8.0 \times 10^{-36}$ | $1.6 \times 10^{-23}$ | $4.8 \times 10^{-42}$ |

For all aptamers, the designed library was enriched in low gene-regulatory activity sequences compared to that aptamer's random library. However, the designed and random libraries exhibited different distributions of gene-regulatory activities, indicating that certain aptamers are more amenable to producing ribozymes with low basal gene-regulatory activity values as determined by the three principal metrics. For example, the analysis of the folinic acid aptamer libraries indicates that the neural network model was successful in designing ribozymes with low levels of gene-regulatory activity for this aptamer. The designed library had a mean gene-regulatory activity value of log10(GFP/mCherry) = −0.36 vs. 0.62 for the random library. The fraction of ribozymes in the designed library with low gene-regulatory activity was 77% vs. 3% for the random library. Finally, the results of the rank-sum test determined that the two libraries had a 0.0% probability (p=$1.2 \times 10^{-91}$) of coming from the same distribution.

Three of the remaining four aptamers (theophylline, tetracycline, chloramphenicol) exhibited similarly successful results in the application of the neural network model for designing ribozymes with low gene-regulatory activities. Specifically, the designed libraries had mean gene-regulatory activity values of 0.31 (theophylline), 0.30 (tetracycline), and 0.38 (chloramphenicol) vs. 0.55, 0.44, and 0.49, respectively, for the corresponding random libraries. The fraction of ribozymes in the designed libraries with low gene-regulatory activity was 15% (theophylline), 8% (tetracycline), and 0.6% (chloramphenicol) vs. 2%, 1%, and 0.1%, respectively, for the corresponding random libraries. While for this set of aptamers the design libraries exhibited more overlap with the corresponding random libraries than the folinic acid libraries in terms of activity, the results of the rank-sum test similarly showed a very low probability of the two libraries for each aptamer coming from the same distribution (theophylline: p=$1.0 \times 10^{-34}$; tetracycline: p=$8.0 \times 10^{-36}$; chloramphenicol: p=$1.6 \times 10^{-23}$). Taken together, the model was successful in each case of designing ribozymes that exhibit low gene-regulatory activities and enriched for such sequences relative to a random library.

Of the five aptamers examined, designing low gene-regulatory activity sequences for the neomycin aptamer proved particularly difficult. The designed neomycin aptamer library had a mean gene-regulatory activity value of log10(GFP/mCherry) = 0.64 vs. 0.73 for the random library. However, the fraction of ribozymes in the designed library with low gene-regulatory activity was 0% vs. 0.01% for the random library. Finally, the results of the rank-sum test determined that the two libraries had a very low probability of coming from the same distribution (p=$4.8 \times 10^{-42}$). The results indicate that while the machine learning model was able to enrich for sequences exhibiting lower gene-regulatory activity relative to a random library, it was not successful in designing sequences in the lower third of the activity range for this aptamer.

Even with the variability inherent in aptamer sequences, the data indicate that the neural network model robustly identifies variable loops that more frequently exhibit lower basal gene-regulatory activity when the opposite loop is constrained to an aptamer sequence than would be expected by random variable loops. Thus, the model is useful in designing ribozyme switch libraries for subsequent

screening. Although high-throughput screening methods such as FACS-Seq can measure many thousands of ribozyme sequences in a single experiment, bottlenecks including transformation efficiency of the cell host can limit the number of sequences that can be practically evaluated (*Olsen et al., 2000*; *Xiang et al., 2019*). Thus, the model can be used as a tool to design libraries of lower sequence diversity biased to those sequences that are likely to exhibit low gene-regulatory activities (*Mishra et al., 2008*).

## The neural network model supports forward design of ribozyme switches for diverse aptamers

We next examined the model's ability to perform the forward design of tertiary-interaction ribozyme switches starting with a given aptamer sequence. In particular, rather than designing tailored libraries that can subsequently be screened for functional switches, the prediction accuracy of the model was sufficient to directly test a small number of the returned highest ranked sequences. To examine the forward design capability of the model, we chose the 15 ribozyme sequences for each aptamer that the model predicted to have the lowest gene-regulatory activities in the absence of ligand (i.e., the 'predicted best'). As the model's objective is solely to design ribozyme sequences with low gene-regulatory activity, inherent in using it to design tertiary-interaction ribozyme switches is the assumption that once such a ribozyme sequence is achieved with an aptamer in loop I or II, binding of the ligand to the aptamer will disrupt the loop interactions and thus the ability of that ribozyme sequence to adopt a catalytically active state at physiological conditions.

Each identified ribozyme sequence was individually synthesized and cloned into the previously described switch characterization plasmid in the 3´ UTR of the GFP reporter. Flow cytometry was used to measure the GFP to mCherry ratio of a clonal cell population harboring a switch characterization plasmid under standard growth conditions and in the absence and presence of high concentrations (1–5 mM) of the cognate ligand for the aptamer (*Figure 6—source data 1*). Gene-regulatory activities were normalized to an inactive control ribozyme to set the theoretical upper limit on activity levels as 100% (*Figure 6—source data 1*).

The model performs consistently well in predicting variable loops that will pair with the folinic acid aptamer to result in very low gene-regulatory activity. Similar to the aptamer's designed library, the top 15 ribozymes designed with the folinic acid aptamer exhibited very low basal gene-regulatory activities, with a median of 3.2% of the nonfunctional ribozyme and a very low interquartile range (the difference between the 75th and 25th percentiles) of 1.8% (*Figure 6A*). In the presence of high concentrations of folinic acid, the median activity increased to 4.7%, resulting in a median activation ratio of 1.5 (*Figure 6B*). The top three performing ribozymes exhibited activation ratios spanning 2.2–2.9 (*Figure 6C*). While two of the top three switches by activation ratio had low basal gene-regulatory activities (2.5% and 4.0%), the third exhibited the highest basal gene-regulatory activity (23.4%) of the top 15 predicted ribozymes. Within the top 15 ribozymes, the sequences that displayed an increase in activity in the presence of folinic acid had the aptamer located in loop I of the ribozyme. Although the model correctly predicted ribozymes exhibiting low gene-regulatory activities with the aptamer located in loop II, that particular architecture either did not allow for ligand binding to the aptamer or for ligand-dependent change in ribozyme cleavage. Overall, the folinic acid-responsive ribozymes did not exhibit large increases in gene-regulatory activities in response to ligand, even though the activities in the absence of ligand were very low. Given the relatively high affinity of the aptamer to its ligand (*Chang, 2014*), these results imply that folinic acid does not cross the cell membrane efficiently, thus resulting in very low intracellular concentrations (*Xiang et al., 2019*; *Niethammer and Kleihauer, 1974*), or does not bind to the aptamer in a way that effectively disrupts ribozyme cleavage.

The model also performs consistently well in predicting variable loops that will pair with the tetracycline aptamer to result in low gene-regulatory activity. The top 15 ribozymes designed with the tetracycline aptamer exhibited low basal gene-regulatory activities, with a median of 26.3% of the nonfunctional ribozyme. There was relatively low variation (interquartile range of 15.8%) in activity across the 15 sequences tested. In the presence of high concentrations of tetracycline, the median activity increased to 75.6%, resulting in a median activation ratio of 2.8. The top three performing ribozymes exhibited activation ratios spanning 4.5–5.8, and all exhibited low basal gene-regulatory activities (spanning 10.7–14.3%). The model predictions for the design of ribozymes with the tetracycline aptamer and folinic acid aptamer provide a notable contrast in that in both instances the

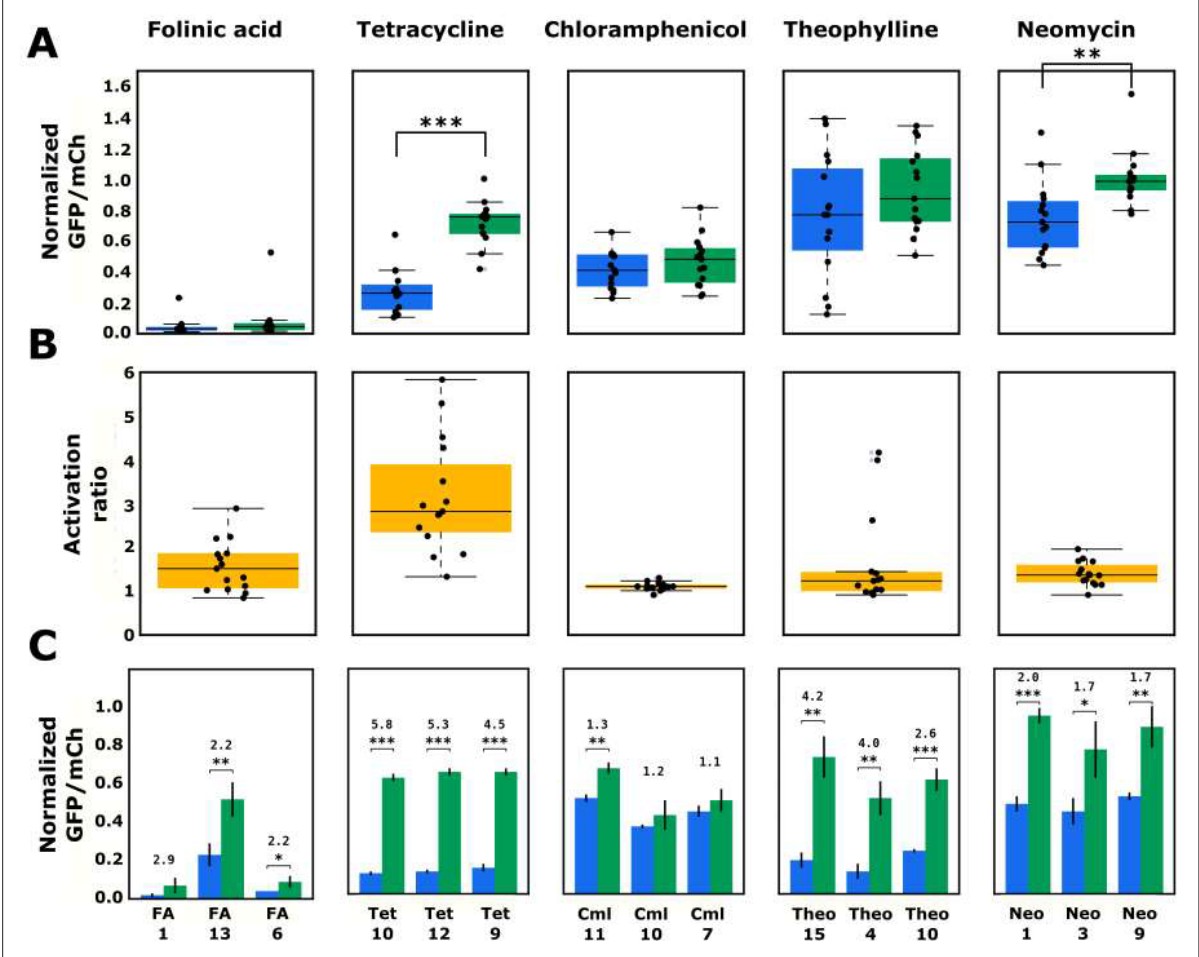

**Figure 6.** Validated basal and activated gene-regulatory activities for the top-performing ribozyme sequences predicted from the convolutional neural network model. (**A**) The basal and activated gene-regulatory activities for the top 15 predicted ribozyme sequences characterized individually via flow cytometry analysis. The box plots show the middle two quartiles of the gene-regulatory activities of ribozymes in the absence (blue) and presence (green) of the indicated ligand, with the mean activity levels for clonal populations of the individual ribozymes represented by the black points. The whiskers extending from the boxes represent 1.5 times the interquartile range, which approximates 99% of the expected range for values. Both basal and activated activity levels are normalized to that of an inactive ribozyme control (sTRSVc). (**B**) The activation ratio values (ratio of activity levels in the presence and absence of the indicated ligand) for the top 15 ribozyme sequences. The box plot represents the middle two quartiles for activation ratio, with the activation ratios for individual sequences represented by the black points. (**C**) The basal and activated gene-regulatory activities and the activation ratio values for the three best performing designs among the top 15 predicted designs. The blue and green bars represent the mean basal and induced gene-regulatory activities, respectively, for an individual ribozyme sequence, with the error bars showing the standard deviation. The number above each pair of bars represents the activation ratio for that sequence. p-values < 0.05 are represented by *, those <0.01 by **, and those <0.001 by ***.

The online version of this article includes the following source data for figure 6:

**Source data 1.** Activity of designed ribozymes in the presence and absence of their cognate ligand.

model reliably predicted sequences with low gene-regulatory activities. The greater success of the tetracycline-responsive ribozymes as genetic switches relative to the folinic acid-responsive ribozymes likely results from particular features of the ligand-aptamer pair, including the ability of tetracycline to more efficiently cross the cell membrane (***Argast and Beck, 1984***).

The model designs with the chloramphenicol aptamer further illustrate that features specific to the ligand-aptamer pair can limit the resulting switch performance, independent of the model's ability to predict variable loop sequences that lead to low gene-regulatory activities. The top 15 ribozymes designed with the chloramphenicol aptamer exhibited low basal gene-regulatory activity with a median of 41.9% of the nonfunctional ribozyme, similar to that of the ribozymes designed with the tetracycline aptamer. In addition, there was relatively low variation (interquartile range of 20.7%) in

activity across the 15 sequences tested. Unlike the tetracycline-responsive ribozymes, the median gene-regulatory activity only increases slightly to 46.1% in the presence of high concentrations of chloramphenicol, resulting in a median activation ratio of 1.1. Among the top three performing ribozyme sequences was one that exhibited a statistically significant activation ratio of 1.3, although its basal gene-regulatory activity was higher than the median chloramphenicol ribozyme basal activity (51.3%). The data demonstrate that the model is able to predict a ribozyme sequence with the chloramphenicol aptamer that exhibits relatively low gene-regulatory activity; however, these sequences exhibited little response to the cognate ligand and thus limited function as genetic switches. Similar to the folinic acid ligand-aptamer pair, these results imply that chloramphenicol does not cross the yeast membrane efficiently (**Abdel-Sayed, 1987**), its toxicity at high concentrations limits the intracellular concentrations attainable, or it does not bind to the aptamer in a way that effectively disrupts ribozyme cleavage.

The model designs with the theophylline aptamer highlight that certain aptamers result in greater variability in the model's ability to predict variable loops that pair with that aptamer to result in low gene-regulatory activity. Although the model produced several ribozymes with low gene-regulatory activities, on average the ribozymes among the predicted top 15 sequences exhibited high gene-regulatory activities in the absence of theophylline, with a median of 77.0% of the nonfunctional ribozyme. In addition, there was substantial variability in the distribution of gene-regulatory activities (interquartile range of 28.1%) across the 15 tested sequences, which was the highest amongst the aptamers tested. In the presence of high concentrations of theophylline, the median activity increased to 80.6%, resulting in a median activation ratio of 1.2. However, due to the variability in activities across the top 15 predicted sequences, the top three performing ribozyme switches exhibited basal gene-regulatory activities as low as 12.7% and activation ratios up to 4.0. Overall, limitations in ligand permeability across the cell membrane or ability of ligand binding to disrupt ribozyme cleavage did not appear to restrict the design of theophylline-responsive ribozymes to the degree observed with some of the other ligands.

Finally, the model's inability to predict variable loops that will pair with the neomycin aptamer to result in low gene-regulatory activity limited the successful design of high-performing switches with this aptamer. Similar to the aptamer's designed library, the top 15 ribozymes designed with the neomycin aptamer did not exhibit low gene-regulatory activities, with a median of 72.5% of the nonfunctional ribozyme. There was relatively large variation in activity across the 15 sequences tested, with an interquartile range of 29.2%. Despite the high gene-regulatory activities in the absence of ligand, a substantial increase in activities was observed in the presence of neomycin, with a median of 98.7%, resulting in a median activation ratio of 1.4. The top three performing sequences exhibited basal gene-regulatory activities as low as 44.3% and activation ratios up to 2.0, thus reaching near the theoretical upper limit of gene-regulatory activity in the presence of ligand (i.e., 100%). The data demonstrate that similarly to tetracycline the neomycin ligand-aptamer pair exhibits properties desirable in the design of genetic switches, particularly the availability of the ligand at intracellular concentrations sufficient to disrupt ribozyme cleavage upon binding its aptamer. While overall the median activities of the sequences predicted by the model for the neomycin aptamer are similar to those predicted for the theophylline aptamer, the model did predict two low gene-regulatory activity sequences for theophylline in the top 15, which was not similarly achieved for neomycin. These results indicate that features of the neomycin aptamer make the model less able to predict loop sequences that will pair with the aptamer to result in ribozyme sequences exhibiting low gene-regulatory activities.

Taken together, the results indicate that the neural network model can be used as a tool for the forward design of individual tertiary-interaction ribozyme switches, eliminating the need for more time-consuming methods like ad hoc design or high-throughput screening. Overall, the design tool produced ribozyme sequences with basal gene-regulatory activities below 33% (or at a level to enable an activation ratio of >3 for aptamer-ligand pairs with optimal features for genetic switches) for four of the five aptamers examined. In addition, the model's predictions resulted in genetic switches exhibiting statistically significant activation ratios for all aptamers examined. While certain aptamers resulted in consistent predictions from the model, others resulted in greater variability. The results from applying the model to these five aptamers support that, in general, testing 15 of the top sequence predictions is sufficient to produce ribozyme sequences with low gene-regulatory activities for those aptamers

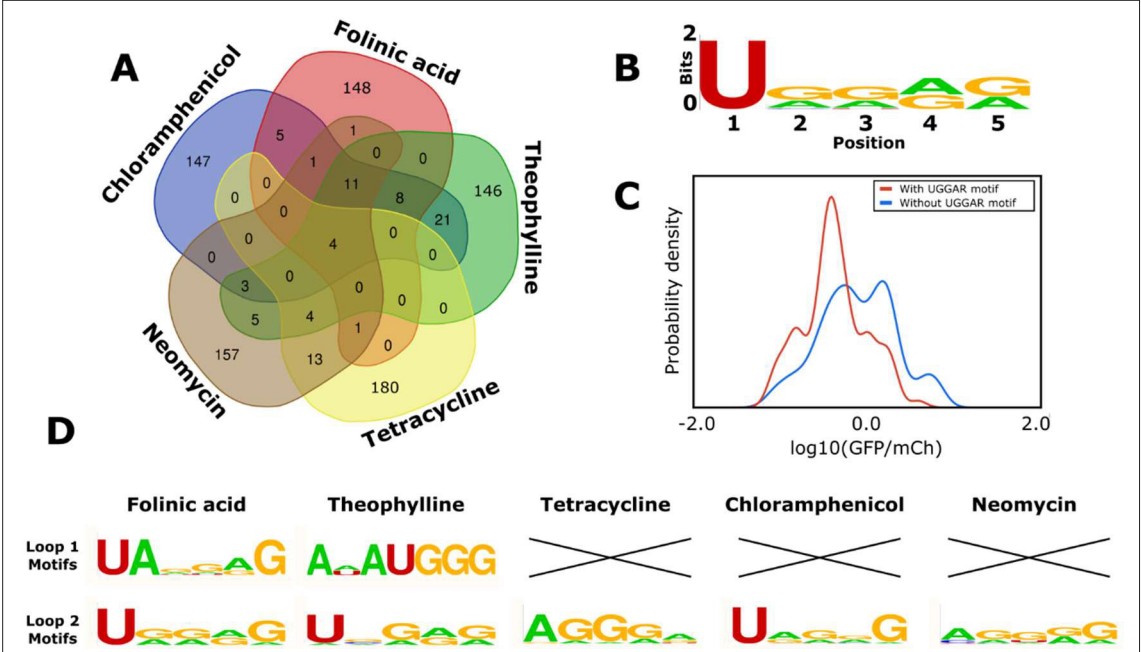

**Figure 7.** Computational analysis of the loop motifs predicted by the convolutional neural network model for different aptamer sequences. (**A**) Venn diagram showing which sequences for the variable loop were shared among the top 200 designs across different aptamers (red: folinic acid; green: theophylline; yellow: tetracycline; brown: neomycin; blue: chloramphenicol). (**B**) Consensus sequence motif for sequences that were present in at least four of the five designed libraries. (**C**) Distribution of expression values for all the sequences in the training library (blue) and sequences harboring the common consensus sequence motif (red). (**D**) Consensus sequence motifs present in each variable loop within the top 15 predicted ribozyme designs for each aptamer. X indicates that there were no designs in the top 15 predicted designs that had that loop present as the variable loop for the indicated aptamer.

whose sequences are intrinsically amenable to creating cleaving ribozymes. The results also show that while designing ribozymes with low basal gene-regulatory activities is an important aspect for generating a switch, there are features specific to each aptamer-ligand pair that can affect switch performance and are not captured by the model.

## Model predictions provide insight into motifs that are enriched in low gene-regulatory activity ribozymes

Finally, we examined the model predictions to determine any trends that provide insight into the sequence-function relationship of the hammerhead ribozyme's loops. We analyzed the sequences of the variable loops in the designed libraries for each aptamer to determine if the model was designing sequences that were specifically optimized for each aptamer or settled on a conserved motif that generally works well and applied that across diverse aptamer sequences. We determined which variable loop sequences appeared across different aptamers (*Figure 7A*). The results of this analysis indicate that 1.5% of the variable loop sequences in the designed libraries are generally favored by the model independent of the aptamer sequence. Specifically, 4 sequences were in every aptamer's designed library (of 200 sequences) and a further 11 sequences were in four of the five libraries. These sequences share a 5-nucleotide motif on loop II (UGGAG) that the model deemed generally favorable for cleavage (*Figure 7B*). We searched in our training set for ribozymes with UGGAG on loop II and a random loop I, and the data indicate that harboring this sequence greatly increases a ribozyme's probability that it will exhibit low gene-regulatory activity levels (*Figure 7C*). Sequences with this motif displayed a mean log10(GFP/mCherry) value of –0.27 vs. a mean log10(GFP/mCherry) value of 0.02 for ribozymes with random loop sequences. Thus, this analysis indicates that the model learned from the training data that this motif is generally likely to result in low gene-regulatory activity levels. In addition, the designed libraries indicate that aptamers may be preferentially located on specific loops to achieve a ribozyme exhibiting low gene-regulatory activities. Specifically, 75% of the folinic acid and 51% of the theophylline designs have the aptamer on loop II, while 83% of the tetracycline, 100%

of the neomycin, and 100% of the chloramphenicol designs have the aptamer on loop I. The designed libraries also demonstrate that the preferred variable loop size varies by aptamer. The neomycin and chloramphenicol aptamers show a strong preference for loop lengths of five or six (91 and 100% of designs, respectively), whereas the tetracycline aptamer shows a strong preference for longer loop lengths of seven or eight nucleotides (85% of designs). In contrast, the folinic acid and theophylline aptamers are distributed equally across all loop lengths.

Though certain motifs were predicted independent of the aptamer sequence, roughly 75% of the sequences in each aptamer's designed library were unique to that library. The lack of a shared motif for the variable loop regions across different aptamers indicates that the model is responding to specific features of the aptamer. The analysis also indicates that the predicted 15 best sequences for each aptamer had different motifs represented (*Figure 7D*). The UGGAR motif that was shared between the designed libraries for multiple aptamers was present in the top designs for the folinic acid, theophylline, and chloramphenicol aptamers and produced switches with basal gene-regulatory activities below 30% for each of these aptamers. The top performing switches for folinic acid and theophylline included sequences with the aptamer on loop II, with different motifs (UANNNG and ANAUGGG, respectively). The top designs for tetracycline and neomycin had different motifs that were enriched in guanine and adenine nucleotides (AGGGA and AGGGG, respectively). One commonality among the various motifs present in the top predicted sequences for the aptamers tested is an enrichment of purine nucleotides, adenine (34.5%) and guanine (49.3%), when compared with the pyrimidines uracil (13.6%) and cytosine (2.6%).

Overall, the area of machine learning concerned with interrogating and explaining neural networks, especially neural networks as applied to genomics, is still new, but as it develops, one could begin applying lessons from that field to explain what contributes to predicted ribozyme activity. Because our model appears to be an accurate predictor of said activity, it could also provide further insights into the biophysical mechanisms governing ribozyme activity.

## Discussion

We developed a computational model that can predict a ribozyme's activity when given the sequence of its two loops and applied that model to the successful design of a set of new ribozymes that can function as genetic switches through the integration of existing aptamer sequences. The neural network model was trained on a data set of hundreds of thousands of ribozyme sequences and associated gene-regulatory activities generated through a FACS-Seq process and was validated to be more accurate and robust to predicting activities of novel structures than simple regression models. We applied the model to the forward design of ribozymes with five aptamer sequences, and in each case the model predictions were more likely to exhibit low basal gene-regulatory activity than ribozymes with random loop sequences. For four of the five aptamers, the model designed ribozymes with basal gene-regulatory activities sufficiently low to enable the design of switches with activation ratios greater than 3.

The model's objective, as developed in this work, is to design ribozyme sequences with low gene-regulatory activity (set at a threshold of less than 30% of the activity of an inactive ribozyme control). Thus, inherent in using the model to design tertiary-interaction ribozyme switches is the assumption that once such a ribozyme sequence is achieved with an aptamer in loop I or II, binding of the ligand to the aptamer will disrupt the loop interactions and thus the ability of that ribozyme sequence to adopt a catalytically active state at physiological conditions. Our results indicate that while the model is generally successful at achieving its stated objective, there are properties specific to aptamer-ligand pairs that make achieving a functional ribozyme sequence with an aptamer integrated in loop I or II not sufficient in all cases to generating a sequence that can act as a ligand-responsive genetic switch. Certain ligands may not be well-suited to being used as effectors of genetic switches used in cells, for example, due to cellular toxicity or membrane permeability of the ligand (*Xiang et al., 2019*). In addition, the structural features of the aptamer-ligand pair, including the nature and location of the binding interactions, may result in ribozyme sequences in which binding of the ligand to the aptamer does not disrupt the tertiary interactions between loops I and II required for catalytic activity at physiological conditions. As a separate consideration, structural features of the aptamer may result in loss or reduction of the binding affinity of that aptamer to its cognate ligand in the context of the full ribozyme sequence (*Xiang et al., 2019*).

To improve the robustness of the model to designing ribozyme switches, features specific to the aptamer-ligand pair that may impact performance of the ribozyme sequence as a switch can be prescreened to identify aptamer-ligand pairs ideally suited for a particular application. For example, knowledge of how or where the ligand binds to the aptamer can inform design choices. As our knowledge and ability to model ligand-aptamer interactions improve, this type of information can be incorporated into future iterations of this design tool. Ligands can be prescreened for low cellular toxicity and high membrane permeability, and in certain cases transporters can be overexpressed to increase intracellular ligand concentrations (*Xiang et al., 2019*). To ensure that integration into the ribozyme does not impact the aptamer's binding affinity, the integrated ribozyme sequence can be prescreened for maintenance of binding affinity using previously described strategies (*Xiang et al., 2019*). As more data is gathered on ribozyme sequence, activity, and switching capability in the future through continued application of high-throughput characterization methods such as FACS-Seq (*Xiang et al., 2019*), that data can be used to train new iterations of the model to identify aptamer structures that are able to generate high-performing ribozyme switches (i.e., achieving low basal gene-regulatory activity and responsiveness to the cognate ligand).

In addition, the encoding of the ribozyme sequence and structure for input into the model is predicated on a assumed 3D structure of the folded ribozyme, especially in regards to the alignment of the two loops, but this alignment is based on the only publicly available 3D structure for the sTRSV hammerhead ribozyme and therefore presents the standard by which to build an assumed structure around. The results demonstrate that there is value in using this assumed structure as the CNN model that used the structural information vastly outperformed the other models that did not have this structural information. Without accurate, predictive 3D modeling algorithms for RNA, the best existing information to build a structure-based model around is the assumed structure. As the 3D modeling algorithms improve, they can be incorporated into future versions of this design tool.

In addition to being useful as a CAD tool for biology, a trained machine learning model can reveal insights into the underlying system it is predicting. We used the results from our model to examine nucleotide interactions between the ribozyme loops that the model believes to be favorable for cleavage at physiological conditions. A previous study that examined sequence diversity across natural hammerhead ribozymes observed a slight preference for purines (*Hammann et al., 2012*) in line with our observations. Thus, future work focusing on more detailed structural analysis of natural and engineered hammerhead ribozyme variants may elucidate the role of purines in the loop regions.

More broadly, our work provides an example of the encoding and machine learning methods that can be used for the prediction of the function of other 3D biological macromolecules including proteins and diverse functional RNAs. It is important that when encoding the system for input into the machine learning model the structure is maintained as much as possible in addition to sequence (*Torng and Altman, 2017*). This will be especially relevant in the future as protein and nucleic acid tertiary structure prediction advances, and one can infer exactly where each amino acid or nucleotide resides in relation to each other. For example, some recent work in applying machine learning to prediction of protein function has shown that the one-hot encoding method classically applied to DNA sequences can be improved upon by splitting the protein's sequence into 'words', or k-mers, of three amino acids and applying natural language processing, which can relate the k-mer to distant sections of the protein (*Yang et al., 2018*). A similar study identifying long noncoding RNA sequences showed that though a k-mer approach has some predictive power, augmenting it with structural information improved the model's accuracy (*Sreeshma et al., 2018*). These examples, along with the work described here, support that machine learning approaches that incorporate structural information on macromolecules outperform approaches that solely use the primary sequence of such macromolecules.

We have made the design tool described here available to the broader research community to aid in the design of genetic switches for diverse applications of interest, through the release of the trained models, code for the high-throughput folding and analysis of ribozymes, and instructions for use along with publicly available tools for the design of novel ribozyme switches ( https://github.com/smolkelab/Ribozyme_switch_generator (*Schmidt and Smolke, 2021*; copy archived at swh:1:rev:-786f72a93791e6d2e7bd35b2bb1ee66ed21a1680). The software is designed to run on low computational resources, and all required external software is free and open source.

As noted in *Figure 3—figure supplement 1*, there is a difference in the error produced by the predictive model when it is evaluated on the training set of ribozymes as compared to a sample test

set. In any complex model with a significant number of features, the model will develop some bias towards the training set, but it is important to attempt to understand where that bias comes from and what the potential ramifications are. We explored the reasons behind this to show that a significant source of bias comes from differences in the structures of ribozymes in the test set vs. ribozymes in the training set. Our training set of ribozymes partially rectifies this by including ribozymes with a diverse set of structures, but potential sources of bias remain that may have contributed to the model being unable to successfully design switches for all five aptamers examined in this work.

Now that libraries of ribozyme switches have been generated that include complex aptamer structures as loops, they could be also included in the training set for the future design of ribozymes with other aptamers and may increase the accuracy by further representing the influence of complex loop structures. Our work describes a design tool that can be used to engineer conditional gene regulation systems that will function in a broad array of cell types including mammalian (*Chen et al., 2010*), yeast (*Townshend et al., 2015*), and bacterial (*Wieland and Hartig, 2008*) cells. The design tool has been demonstrated for the forward design of tertiary-interaction ribozyme switches, which previously were generated via time-intensive cell-based screens of large libraries to identify high-performing switches (*Townshend et al., 2015*). More broadly, computational design tools for biological components will play a key role in advancing our ability to engineer biological systems with programmed functions.

# Materials and methods

## Key resources table

| Reagent type (species) or resource | Designation | Source or reference | Identifiers | Additional information |
|---|---|---|---|---|
| Strain, strain background (*Escherichia coli*) | DH10B | Thermo Fisher | C404010 | Chemically competent |
| Strain, strain background (*Saccharomyces cerevisiae*) | W303 *MATα* | https://www.yeastgenome.org/strain/w303 | | Chemically and electrocompetent |
| Recombinant DNA reagent | pCS1748 | PMID:22810204 | | Used to characterize ribozyme activity |
| Sequence-based reagent | PCR and sequencing primers | This paper | PCR primers | Sequences listed in supplementary information |
| Commercial assay or kit | KAPA HiFi PCR Kit | Roche Sequencing Store | KR0368 | |
| Chemical compound, drug | Folinic acid | Millipore Sigma | F7878 | |
| Chemical compound, drug | Theophylline | Sigma Aldrich | T1633 | |
| Chemical compound, drug | Neomycin | Sigma Aldrich | N1876 | |
| Chemical compound, drug | Chloramphenicol | Sigma Aldrich | C0378 | |
| Chemical compound, drug | Tetracycline | Sigma Aldrich | 87128 | |
| Software, algorithm | Python 3 | Python.org | | |
| Software, algorithm | RNAstructure | *Reuter and Mathews, 2010* | | |

Oligonucleotides used in generating plasmids and barcoding for sequencing appear in *Supplementary files 1–4*. All yeast used in this work were of the CSY3 strain (W303 *MATα*). All *E. coli* used for cloning were of the Top10 strain. KAPA HiFi PCR Kit (Kapa Biosystems) was used for PCR amplifications according to the manufacturer's instructions, unless described otherwise. Sanger sequencing was performed by Elim Biopharmaceuticals, Inc. The code for running computational methods reported here can be accessed at https://github.com/smolkelab/Ribozyme_switch_generator (*Schmidt and Smolke, 2021*).

## Construction of ribozyme libraries with randomized loops and aptamer loops for use as a training set

For the ribozyme libraries, oligonucleotides were ordered that contained N5−N14 in place of the two loops (*Supplementary file 1*). The oligonucleotides were designed such that the forward oligonucleotides contained the first half of the ribozyme, from the 5′ spacer sequence through the catalytic core, including the random library for loop I. The reverse oligonucleotides contained the catalytic

core through the 3′ spacer sequence, including the random library for loop II. The overlap in the catalytic core area allowed for annealing during the PCR assembly. The spacer sequences were designed to minimize interactions between the surrounding sequences and the switch sequence. The switch libraries were constructed through a one-pot PCR reaction using the KAPA HiFi PCR Kit (Kapa Biosystems). For the PCR, 10 cycles were performed with a melting step of 94°C for 30 s, an annealing step of 53°C for 30 s, and an extension step of 72°C for 15 s. This was followed by 20 cycles with a melting step of 94°C for 30 s, an annealing step of 65°C for 30 s, and an extension step of 72°C for 15 s. For library 1, both the forward and reverse oligos containing N5–N10 were used in one reaction. For library 2, two reactions were prepared – one with forward primers containing N5–N7 and reverse primers containing N9–N14 and one with forward primers containing N9–N14 and reverse primers containing N5–N7 (*Supplementary file 1*). These mixtures were amplified with the primers Gaprepair_F and Gaprepair_R (*Supplementary file 1*).

## Cloning and yeast transformation of ribozyme libraries

The assembled ribozyme libraries were cloned via homologous recombination into the plasmid backbone pCS1748 via electroporation in yeast. Briefly, 50 mL yeast culture (OD600 = 1.3–1.5) was incubated with Tris-DTT buffer (2.5 M DTT, 1 M Tris, pH 8.0) for 20 min at 30°C, pelleted, washed, and resuspended in buffer E (10 mM Tris, pH 7.5, 2 mM MgCl$_2$) to 250 μL. To 60 μL of the yeast cell suspension, 2 μg of linearized plasmid and 1 μg of library insert DNA were added, and the DNA-cell suspension was electroporated (2 mm gap cuvette, 540 V, 25 μF, 1000 Ω). Transformed cells were diluted to 1 mL volume in yeast peptone dextrose medium and incubated for 1 hr at 30°C. The incubated cultures were spun down and resuspended in 100 μL of water, which was then transferred directly into 50 mL YNB-URA liquid media for growth in 30°C. 100 μL of a 1/10,000 dilution of each transformation was also plated on YNB-URA agar plates to obtain the transformation efficiencies. Six transformations were in parallel done for each library to ensure sufficient coverage.

## Fluorescence-activated cell sorting of ribozyme libraries

Every 12 hr, the cells harboring the ribozyme libraries were back-diluted into fresh 100 mL YNB-URA media and grown at 30°C to ensure that cells remained in exponential phase. This was repeated for six re-passagings. Replicate transformations were combined or discarded once the transformation efficiencies were determined so that the number of transformed cells reached an appropriate amount, generally 10× the targeted number of sequences. The libraries were passaged for 3–4 days prior to sorting.

For cell sorting, the passaged library was back-diluted to an OD of 0.1 in two separate 100 mL YNB-URA media flasks. The two cultures served as technical replicates, ensuring that the expression, sorting, and sequencing did not introduce significant variability in the measurement of ribozyme gene-regulatory activity levels. The two cultures were grown for 7 hr, then spun down at 600 rpg and resuspended in 2 mL phosphate buffered saline (PBS) (Life Technologies) + 1 % Bovine serum albumin (BSA). The resuspended cultures were kept on ice until sorting.

Sorting was performed on a FACSAria II FACS machine (Beckman Coulter) at the Stanford FACS Core. 8 bins were generated using a Matlab script (https://github.com/btownshend/TwoColor, *Townshend, 2018*), which ensured that each bin except the first and last spanned the same log10(GFP/mCherry) distance. GFP was excited at 488 nm and measured with a splitter of 505 nm and bandpass filter of 525/50 nm. mCherry was excited at 532 nm and measured with a splitter of 600 nm and bandpass filter of 610/20 nm. Initial gates based on the forward-scatter area and side-scatter area were used to gate out cell debris and nonviable cells. The bins were set so that the upper bin and the lower bin each contained roughly the top and bottom percentile of expression levels, respectively, with the remaining six bins capturing the remaining levels of expression. 20–25 million cells from each of the replicate cultures were sorted into the various bins based on the ratio between GFP and mCherry expression. Cells falling into bins 1–4 were collected in the first sort, and all other cells were discarded. In the second sort, cells falling into bins 5–8 were collected. 5 mL YNB-URA was added to each bin from the sort, though more YNB was added to some bins so that the ratio was at least 1:2 PBS:YNB. The cultures for the recovered bins were grown up overnight and 1 mL of each bin was miniprepped the next day using the ZR Fungal/Bacterial DNA MiniPrep kit (Zymo Research) according

to the manufacturer's instructions. The rest of the culture from each recovered bin was spun down and frozen as backup.

## Prescreening of ribozyme libraries

After several days of post-transformation passaging, certain experiments performed a pre-sort on the library to achieve a desired frequency of 'low-expressing' cells. In these pre-sorts, cells harboring the libraries were sorted into low-expression and high-expression groups as appropriate. The gates for sorting were set so that a certain bottom percentile of expressing cells were designated as 'low-expressing' while the top percentile of expressing cells were 'high-expressing'. For library 2, the low-expressing cells were defined as those in the bottom 20% of the entire library 2 population. For library 3, the low-expressing cells were defined as those in the bottom 5% of the entire library 3 population. Post-sorting, the sorted cell groups were passaged into fresh 100 mL YNB-URA media and maintained for 2 days before the final eight-bin sort. If the low- and high-expressing populations were both included in the final sort, as in library 2, they were mixed at the point of the back-dilution before the final 7 hr of growth so that the appropriate ratio of low- and high-expressing cells was obtained.

## NGS sample preparation and sequencing

Miniprepped DNA from the binned sorts were first PCR-amplified using primers GFP_F and ADH1t_R (*Supplementary file 2*) and the KAPA HiFi PCR Kit (Kapa Biosystems) to isolate from the miniprepped DNA the portion of the plasmid that contained the ribozyme library. For the PCR, 15 cycles were performed with a melting step of 94°C for 30 s, an annealing step of 57°C for 30 s, and an extension step of 72°C for 15 s. The resulting PCR products were barcoded using the primers listed in *Supplementary file 2*, with each bin PCR-amplified using a different combination of forward and reverse primers that added different 5′ and 3′ barcodes. For the PCR, 15 cycles were performed with a melting step of 94°C for 30 s, an annealing step of 57°C for 30 s, and an extension step of 72°C for 15 s. The forward primer contained a barcode indicating whether the bin contained the induced or uninduced library (BT1334 and BT1335), while the reverse primer contained a barcode indicating which bin was being amplified (BT609–BT1462). For the PCR, 15 cycles were performed with a melting step of 94°C for 30 s, an annealing step of 57°C for 30 s, and an extension step of 72°C for 15 s. The relative concentrations of each amplified bin were determined using qPCR with primers BT235 and BT236 and Fast Plus EvaGreen qPCR master mix (Biotium). Each amplified bin was tested at 1/100, 1/400, 1/1000, and 1/4000 dilutions. The concentration measured for these four dilutions was averaged together to determine the concentration for that bin. The amplified bins were mixed together so that the final mixture would contain a fraction of molecules from each bin in proportion to the number of cells that were originally sorted into that bin. For example, if 50 million total cells were sorted into bins during the FACS portion and 5 million of those were sorted into bin 3, then the number of molecules from the amplified bin 3 made up 10% of the final mixture. The final mixture was PCR-amplified using primers BT235 and BT236 (*Supplementary file 2*) to increase the concentration. Samples were quantified on both a Bioanalyzer 2100 (Agilent Technologies) and a Qubit 3.0 Fluorometer (Thermo Fisher Scientific). The resulting product was sequenced using a MiSeq machine (Illumina) at the Stanford PAN Facility with a 2 × 150 kit. Samples were run using Illumina standard procedures, with PhiX (Illumina) added (to 15% by molarity) to further increase diversity at nucleotide positions that would otherwise have a significant fraction of the sample sharing the same base call and result in lower read quality.

## NGS data processing

NGS reads were paired using the paired-end read merge program PEAR (*Zhang et al., 2014*). The paired reads were analyzed by a Python script that recorded for each sequence the number of times that sequence appeared in the reads from each bin, normalized by the total number of reads that were obtained from that bin. A normal distribution was fitted to the normalized number of reads for each bin to obtain a mean GFP/mCherry value. The fits were performed using custom Python code available at https://github.com/smolkelab/Ribozyme_switch_generator. The fits resulted in an estimate for each sequence of the GFP/mCherry ratio.

## Computational prediction of ribozyme secondary structure

We used the RNA secondary structure prediction software RNAstructure (*Reuter and Mathews, 2010*) for the computational folding of ribozymes as it has good accuracy, with a correlation between predicted and known RNA structures among the highest of all RNA secondary structure prediction methods (*Puton et al., 2013*), on short RNA sequences and can fold several sequences of ribozyme length (roughly 100 nucleotides) per second on a standard desktop computer. To confirm this, we used the stochastic folding tool from RNAstructure, which gives a distribution of predicted structures, on each of the designed ribozyme switches for the five aptamers that were predicted to have the lowest basal gene-regulatory activity. For each of these ribozyme switches, we gathered a distribution of 100 predicted structures and did not observe an alternate structure among this test set. We have included this data in the supplementary information (*Supplementary file 5*). In order to facilitate the high-throughput folding of hundreds of thousands of ribozymes, we use a Python script to control RNAstructure's command line interface and collect the minimal free energy structure for each sequence of a list of ribozyme sequences. The Python script then automatically analyzes each ribozyme's structure, pulling out important features such as loop size, loop sequence, and stem length. The library of folded ribozymes is stored as a Python list where each entry is a single ribozyme sequence with its corresponding structure, structural components, and, for ribozymes whose activities have been measured through FACS-Seq, basal and activated log10(GFP/mCherry) values. This list is stored long term as a Python list using the Pickle package. The code for performing the high-throughput computational determination of ribozyme structure can be found at https://github.com/smolkelab/Ribozyme_switch_generator/blob/master/Generate_candidate_list.py and https://github.com/smolkelab/Ribozyme_switch_generator/blob/master/Fold_candidate_list.py.

## Encoding of a ribozyme sequence for machine learning

The data set of ribozymes with known gene-regulatory activities is retrieved from storage by using Pickle to load the list of ribozyme sequences and structures into memory. The ribozymes are then grouped by ribozyme structure into Python dictionaries where all the ribozymes in each dictionary have the same structure. When training and test sets need to be made, the relevant dictionaries are combined. These dictionaries are then put through a function that generates an array from the dictionary that represents the encoded ribozymes. The array generation functions work by iterating through each ribozyme in the input dictionary, analyzing its structure and loop sequence as described. The code for encoding of a ribozyme's sequence and structure for machine learning can be found at https://github.com/smolkelab/Ribozyme_switch_generator/blob/master/Train_additional_models.py.

## Model training and prediction of ribozyme gene-regulatory activity

To implement a lasso algorithm, we use the built-in function linear_model.LassoCV from the scikit-learn machine learning library (*Pedregosa et al., 2011*). This model is fit to the array generated for the training data using gradient descent and generates an activity prediction based on the test array. We use Google's tensorflow (*Martín, 2015*) to run the CNN using the Keras package for the easy implementation of the described design (keras.io). The model is trained on the training data by using the Adam optimizer (*Kingma, 2014*) to minimize the mean squared error on the training set, iterating over 100 epochs. After training, the model can be used to generate a predicted basal activity for a test array. The code for training a model and using it for prediction can be found at https://github.com/smolkelab/Ribozyme_switch_generator/blob/master/Train_additional_models.py and https://github.com/smolkelab/Ribozyme_switch_generator/blob/master/Predict_activities.py.

## Model-based forward design of ribozymes with one aptamer loop

For a given aptamer, the bulge where the ligand binds was identified through literature. If there was no literature suggesting a ligand-binding area, the bulge closest to the terminal stem was chosen. The stem leading up to this bulge was removed and the remaining nucleotides of the aptamer were attached to either the first or second stem of the ribozyme, therefore completing that loop (*Figure 5—figure supplement 2*). The opposite ribozyme loop was randomized to an N5–N8 sequence. The full set of designed ribozyme switch sequences was put through the machine learning protocol for the computational determination of secondary structure, encoding, and gene-regulatory activity predicting. The 200 sequences with the lowest predicted basal gene-regulatory activity levels were

chosen as the best candidates. This protocol of turning an aptamer into a library of possible ribozyme switches, predicting their basal gene-regulatory activity, and selecting the 200 sequences that are predicted to have the lowest basal gene-regulatory activity was repeated with the five aptamers to theophylline, folinic acid, neomycin, tetracycline, and chloramphenicol, resulting in 1000 candidate sequences (*Supplementary file 6*).

## Designed library construction and yeast transformation

The oligos used for constructing the designed library of variable loops were ordered as an oligo pool (Twist Bioscience) and assembled into a library of switches through PCR. The forward primers contained the 5′ spacer through the first half of stem II, including either the aptamer or the variable loop I. The reverse primers contained the second half of stem I through the 3′ spacer, including either the aptamer or variable loop II (*Supplementary file 4*). Two PCR mixes were made – one mix with forward primers containing aptamers and reverse primers containing variable loop II and one mix with forward primers containing variable loop I and reverse primers containing aptamers. These mixtures were PCR-amplified with the primers switch_GFP_AvrII_Fwd and switch_ADH1t_XhoI_Rev (*Supplementary file 4*). For the PCR, 10 cycles were performed with a melting step of 94°C for 30 s, an annealing step of 53°C for 30 s, and an extension step of 72°C for 15 s. This was followed by 20 cycles with a melting step of 94°C for 30 s, an annealing step of 65°C for 30 s, and an extension step of 72°C for 15 s. In addition to a designed library, a random library was generated by using primers with random sequences of N5–N8 in the place of the variable loop. These primers were used to construct a random library in the same PCR-based protocol through which the designed library was constructed.

The assembled libraries were transformed alongside the digested plasmid backbone pCS1748 into electrocompetent yeast as described previously. Three transformations were performed for each library to ensure sufficient coverage.

## Cloning of designed ribozyme sequences into characterization plasmids

All riboswitch and backbone plasmid sequences are listed in *Supplementary file 4*. Designed ribozymes and ribozyme switches were assembled from primers and amplified through PCR with the primers switch_GFP_AvrII_Fwd and switch_ADH1t_XhoI_Rev (*Supplementary file 4*) using PFU Ultra II HS (Agilent Technologies). For the PCR, 10 cycles were performed with a melting step of 94°C for 30 s, an annealing step of 53°C for 30 s, and an extension step of 72°C for 15 s. This was followed by 20 cycles with a melting step of 94°C for 30 s, an annealing step of 65°C for 30 s, and an extension step of 72°C for 15 s. The plasmid backbone pCS1748, which contains both *yEGFP* and *ymCherry* under the control of two separate *TEF1* promoters, was digested overnight with restriction enzymes AvrII and XhoI. The assembled ribozyme cassette was cloned into the plasmid backbone through a Gibson Reaction (*Gibson et al., 2009*) and transformed into Top10 *E. coli* for recovery and sequence verification.

## Fluorescence measurements of clonal populations through flow cytometry

Individual plasmids with designed ribozymes cloned between *yEGFP* and the *ADH1* terminator were transformed into chemically competent CSY3 using the Frozen-EZ Yeast Transformation II chemical competence kit (Zymo Research) according to the manufacturer's instructions. Three colonies from each transformation were picked into a 96-well plate with 500 µL YNB-URA in each well. Cultures were grown overnight in a 30°C shaker. 10.5 µL of saturated culture was then transferred into 200 µL YNB-URA either with or without the inducing molecule. The concentrations used were 5 mM for folinic acid, 1 mM for tetracycline, 25 µg/mL for chloramphenicol, 5 mM for theophylline, and 5 mM for neomycin. Cultures were grown for 6 hr in a 30°C shaker. The plates were spun at 3500 rpm for 5 min and the supernatant was discarded. The remaining cells in each well were resuspended in 200 µL PBS + 1% BSA and filtered.

Single-cell fluorescence levels were recorded using the VYB flow cytometer (Miltenyi), measuring both GFP and mCherry fluorescence levels for up to 10,000 cells per sample. GFP (the FL7.A channel) was excited at 488 nm and measured with a splitter of 505 nm and bandpass filter of 525/50 nm. mCherry (the FL4.A channel) was excited at 532 nm and measured with a splitter of 600 nm and bandpass filter of 610/20 nm. Initial gates based on the forward-scatter area and

side-scatter area were used to gate out cell debris and nonviable cells. Cells that displayed an FL4.A reading of less than $10^{-3.5}$ were removed from analysis as these cells had lost the mCherry gene through recombination. The normalized expression level (GFP/mCherry) for an individual cell was calculated by dividing the cell's FL7.A reading by its FL4.A reading. For each sample, the GFP/mCherry values of all the cells in the sample were averaged to calculate an overall GFP/mCherry for the sample. For each switch, the samples are averaged together to obtain a mean GFP/mCherry value for that switch.

The GFP/mCherry values of all switches were normalized by dividing the GFP/mCherry value for that switch by the GFP/mCherry value for yeast transformed with a control ribozyme that does not cleave. For each switch, its uninduced GFP/mCherry value was divided by the GFP/mCherry value of an uninduced control ribozyme sample, while the switch's induced GFP/mCherry value was divided by the GFP/mCherry value of an induced control ribozyme sample. The activation ratio for a switch was calculated by dividing the normalized induced GFP/mCherry by the uninduced GFP/mCherry.

## Acknowledgements

We thank Dr Ben Kotopka, Dr James Payne, and Deze Kong for comments during the preparation of the manuscript; Dr Brent Townshend for guidance in developing and executing the FACS-Seq protocols; and Dr Ben Kotopka for expertise in developing and troubleshooting machine learning models. Cell sorting/flow cytometry analysis for this project was done on instruments in the Stanford Shared FACS Facility. This work was supported by funds from the National Institutes of Health (grant to CDS) and the National Institute of Standards and Technology (training grant to CMS).

## Additional information

### Funding

| Funder | Grant reference number | Author |
| --- | --- | --- |
| National Institutes of Health | R01 GM086663 | Christina D Smolke |
| National Institute of Standards and Technology | Training grant | Calvin M Schmidt |

The funders had no role in study design, data collection and interpretation, or the decision to submit the work for publication.

### Author contributions

Calvin M Schmidt, Conceptualization, Formal analysis, Investigation, Methodology, Validation, Visualization, Writing – original draft; Christina D Smolke, Conceptualization, funding-acquisition, Methodology, project-administration, supervision, writing-review-and-editing

### Author ORCIDs

Calvin M Schmidt http://orcid.org/0000-0002-9507-7507
Christina D Smolke http://orcid.org/0000-0002-5449-8495

### Decision letter and Author response

Decision letter https://doi.org/10.7554/eLife.59697.sa1
Author response https://doi.org/10.7554/eLife.59697.sa2

## Additional files

### Supplementary files

• Supplementary file 1. Primers used to create the training libraries in FACS-Seq experiments 1–3. Primers are DNA oligonucleotide sequences.

• Supplementary file 2. DNA primers used in the next-generation sequencing (NGS) preparation for all FACS-Seq experiments. Primers are DNA oligonucleotide sequences.

• Supplementary file 3. DNA primers used in the construction of the ribozyme plus aptamer library in FACS-Seq 4. Primers are DNA oligonucleotide sequences.

• Supplementary file 4. DNA primers used in the construction of the individual designed ribozymes. Primers are DNA oligonucleotide sequences.

• Supplementary file 5. Top-predicted structures of the ribozymes with the lowest predicted basal gene-regulatory activities. Structures are laid out in dot-bracket notation.

• Supplementary file 6. Sequences for loops I and II for the 200 ribozymes for each aptamer that were predicted to have the lowest basal gene-regulatory activity. The sequences of loops I and II are indicated, one of which is the indicated aptamer, while the remainder of the ribozyme sequence is conserved. The final column contains the predicted basal gene-regulatory activity for a ribozyme with the two given loop sequences in terms of log10(GFP/mCherry).

• Transparent reporting form

### Data availability

All data generated or analyzed during this study and including in the manuscript and supporting file. Source data files are provided where appropriate.

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
