## [Decision Letter]

**Acceptance summary:**

This study presents a convolutional neural network (CNN) model for predicting ribozyme activity from primary sequence, based on measured regulatory activities of tens of thousands of ribozymes with diverse sequence features. The model trained on these sequences can be used to predict the regulatory activities of ribozymes containing ligand responsive aptamers (aptazymes).

**Decision letter after peer review:**

Thank you for submitting your article "A convolutional neural network for the prediction and forward design of ribozyme-based gene-control elements" for consideration by *eLife*. Your article has been reviewed by 3 peer reviewers, and the evaluation has been overseen by Detlef Weigel as the Reviewing and Senior Editor. The following individual involved in review of your submission has agreed to reveal their identity: Natarajan Kannan (Reviewer #1).

The reviewers have discussed the reviews with one another and the Reviewing Editor has drafted this decision to help you prepare a revised submission.

Summary:

The authors present attempts to build a convolutional neural network (CNN) model for predicting ribozyme activity from primary sequence. The authors started by measuring the regulatory activities of tens of thousands of ribozymes with diverse sequence features. A model trained on these sequences was used to predict the regulatory activities of ribozymes containing ligand responsive aptamers (aptazymes) for five different ligand:aptamer pairs. Experimental validation allowed for the identification of ribozyme switches for four aptamers.

The reviewers all appreciated the general approach, but raised concerns about the overall performance of the CNN, which might be caused by overfitting in the NN model and lack of ligand-binding experiments in high-throughput experiments. A related weakness that the CNN is not compared to other, more straightforward machine learning methods.

Essential revisions:

1. There is considerable indication that the NN model is overfitting. This is reflected from its dramatic performance drop when applying to the sTRSV hammerhead ribozyme and large variation of R^2^ (0.7 to 0.46) for 4-structure fold cross-validation.

2. The input coding of the NN model assumes that the 3D structures of all the library sequences are the same, which is almost certainly not the case. This is perhaps one of the main reasons for the poor generalization ability of the NN model. Related to this, the encoding of ribozyme structures assumes that LoopI and LoopII are always aligned in the same manner. It is unclear if this is a valid assumption and how this would impact the predictions. A further issue might be the predicted secondary structure, which is only about 70% correct, or that multiple folds are possible for the same RNA sequence. It is unclear how this is encoded in the model and whether aptamers of different folds (Y-loop for example) are represented in the training datasets. Along similar lines, have the authors considered encoding features of the ligand along with aptamer sequences? This might improve overall accuracy and account for the low performance observed for some aptamer ligand pairs such as the neomycin aptamers in the current model.

3. The authors perform a comprehensive comparison of the performance of a lasso model with CNN and show that the CNN model consistently performs better, even on ribozymes with novel loop structures. However, use of the CNN and the sole comparisons with the lasso model are not well justified. The authors should include comparisons with other methods such as SVM, random forest, or perhaps even ensemble methods.

4. Have the authors considered the following explanation for why their method has limited success: If the aim is to increase the activation ratio, the high-throughput experiments should include the experiments in the presence of ligands. Seeking sequences of low gene-regulatory activities and hoping for high activation ratios will not work as this paper shows.

5. The authors use a wide range of loop lengths and features to construct libraries for training and initial testing of their model, but find that their algorithm does not work as well for complex libraries. Did the authors try training their model on complex libraries including the libraries of 174,080 ribozyme sequences paired with aptamers? If not, this might be worth trying out.

Other points:

1. The introduction section and the significance value of the manuscript is primarily focused on RNA switches. Although the goal of the study was to facilitate finding ribozyme switches, the CNN was not very successful at doing so, beyond a few switches. The authors should revisit their narrative to focus on the background and applicability of the major outcome of their tool: finding regulatory ribozymes.

2. While the application of CNN for ribozyme design is very interesting, the "back-box" nature of the models precludes a direct application of these methods for ribozyme engineering and design. From the results, it is unclear what specific sequence and conformational features are contributing to the observed accuracy in predicting ribozyme activity. The authors should expand on the "explainable" aspects of their model to substantiate their design/engineering claims.

3. In the introduction, the authors refer to previously developed software for identifying RNA switches. It is unclear how the machine learning approach described here is better at predicting RNA switches than previously developed software tools pointed out by the authors. In fact, the other softwares identify more switches. The authors should bring out the benefits of using their algorithm over others.

4. By saying "However, none of the computational design tools described to date are based on gene regulatory platforms that function across a broad range of organisms.", it is implied that the current manuscript is also focused on finding broad-range switches. However, the new aptazymes described have been tested only in Yeast. It will make the manuscript better if the authors could test some of the high performing switches in human or other model organisms.

5. The abstract and the end paragraph of introduction mention different number of aptamer switches that were identified. 5 in the abstract and 4 in introduction and later sections.

6. At multiple places in the manuscript the authors attribute the lack of ligand dependent switch responsiveness to impermeability of the cells to specific ligands. There is however no direct evidence for this reasoning. In some cases, these statements also lack references, for example "…including the ability of tetracycline to more efficiently cross the cell membrane."

7. Several parts of the result section describe data without referring to the associated table/Figure.

8. "The method starts by using the RNA secondary structure prediction program RNAstructure (21) to determine the sequence of loops I and II of the ribozyme (Figure 3A)." What exactly is meant, as RNAstructure employs sequences to predict secondary structure?

9. "Next, the sequences of the two loops are analyzed to determine the nucleotides and their positions in the loops". Not clear. If the sequence is known, would their positions in the loops be known?

10. How are stem-loop interactions used as input in the 3-D image? Where does the interaction information come from? Was the input mapping onto the known 3D structure of hammerhead ribozymes? What about other ribozymes, do the authors input their 3D structures as well? Can the method be applied to a ribozyme with unknown 3D structures?

11. Is the average expression for 15 ribozymes shown in Figure 6A significantly different in the presence or absence of ligands? Box plots in Figure 6A needs p-values.

12. Since Figure 6B is a reanalysis of data shown in Fig6A, it can be moved to supplementary. Figure 6A will be more informative if the points were connected between the two box plots so the readers can observe switch in activity for every plotted ribozyme.

13. The high R-squared values (0.91 and 0.82) in Figure 4B are a bit surprising considering the poor fit especially in the head and tail regions. The authors need to check their fit and R-squared calculations. Likewise, R-square values reported in supplementary Figure 1A (0.92) and 1B (R-squared = 0.94) needs to be cross checked for accuracy.

[Editors' note: further revisions were suggested prior to acceptance, as described below.]

Thank you for resubmitting your work entitled "A convolutional neural network for the prediction and forward design of ribozyme-based gene-control elements" for further consideration by *eLife*. Your revised article has been evaluated by Detlef Weigel (Deputy Editor) and two outside experts.

The manuscript has been improved but there is a smallish remaining issue that should be addressed:

Mean squared errors of the test set are still at least three times larger than those of the training set (Figure 3—figure supplement 1). Thus, the method is clearly biased toward the training set and it is not that clear how generalisable it is. It is a bit difficult to understand why this should be an advantage rather than a negative. In any case, please discuss and emphasize in the abstract and in the Discussion section, so that the reader has a clear sense of the limitation of the technique.

---

## [Author Response]

Essential revisions:

1. There is considerable indication that the NN model is overfitting. This is reflected from its dramatic performance drop when applying to the sTRSV hammerhead ribozyme and large variation of R^2^ (0.7 to 0.46) for 4-structure fold cross-validation.

We thank the reviewer for this comment. The values shown in this figure are example values and not real data. The range of actual R^2^ values for this workflow of splitting the measured ribozymes into a test and training set based on structure are presented in Figure 4F. To avoid confusion, we have changed the values in Figure 4E to match the actual values.

However, It is true that the model can overfit to specific structures. It is for this reason that we generated a library of training data with a variety of loop sizes and shapes – so that when one has a novel aptamer that one wants to incorporate into a ribozyme switch there are likely to be structures in the training set that are similar to the structure that occurs when the aptamer is incorporated into the ribozyme. Supplementary Figure 5C further demonstrates the importance of having structures in the training set that are similar to structures in the test set by demonstrating how the R^2^ value deteriorates as the difference in structure increases. We have updated the text to further emphasize the importance of this relationship.

2. The input coding of the NN model assumes that the 3D structures of all the library sequences are the same, which is almost certainly not the case. This is perhaps one of the main reasons for the poor generalization ability of the NN model. Related to this, the encoding of ribozyme structures assumes that LoopI and LoopII are always aligned in the same manner. It is unclear if this is a valid assumption and how this would impact the predictions. A further issue might be the predicted secondary structure, which is only about 70% correct, or that multiple folds are possible for the same RNA sequence. It is unclear how this is encoded in the model and whether aptamers of different folds (Y-loop for example) are represented in the training datasets. Along similar lines, have the authors considered encoding features of the ligand along with aptamer sequences? This might improve overall accuracy and account for the low performance observed for some aptamer ligand pairs such as the neomycin aptamers in the current model.

We thank the reviewer for this comment. It is true that the NN model is based on a assumed 3D structure of the folded ribozyme, especially in regards to the alignment of the 2 loops, but this alignment is based on the only publicly-available 3D structure for the sTRSV hammerhead ribozyme and therefore presents the standard by which to build an assumed structure around. The results demonstrate that there is value in using this assumed structure, as the CNN model that used the structural information vastly outperformed the other models which did not have this structural information. Without accurate, predictive 3D modeling algorithms for RNA, the best existing information to build a structure-based model around is the assumed structure. As the 3D modeling algorithms improve, they can be incorporated into future versions of this design tool. We have updated the text of the manuscript to acknowledge this structural assumption and highlight this point more clearly in the Discussion section.

While the prediction of secondary structure, although more accurate than the prediction of tertiary structure, is still not perfect, one of the benefits of using the hammerhead ribozyme as the base for this design software is that the hammerhead ribozyme has a strong tendency to fold into its canonical secondary structure and avoid alternative structures. To confirm this, we used the stochastic folding tool from RNAstructure, which gives a distribution of predicted structures, on each of the designed ribozyme switches for the 5 aptamers that were predicted to have the lowest basal gene-regulatory activity. For each of these ribozyme switches, we gathered a distribution of 100 predicted structures and did not observe an alternate structure among this test set. We have included this data in the supplementary information (Supplementary Table 7). As mentioned in the text, RNAstructure has good prediction accuracy on RNA molecules of the length of these ribozyme switches, and the software does not predict that multiple folds are common, giving us confidence that the ribozyme switches are generally folding into the MFE structure.

In response to the reviewer’s comment on the secondary structures observed in the training set, we want to clarify that the training set does not in fact contain any aptamers, but does contain loops with random sequences up to 13 nucleotides, which can create stems extending off of the loop. Of the larger loops (length 10 nucleotides or greater), between ⅓ and ½ of the loops in the training set contain a stem branching off of the loop, providing significant structural diversity to our training set. We have updated the manuscript text with information on the structures observed in the training set.

Finally, the reviewer makes an interesting point about encoding other features of the aptamer for use in the model. Aptamer-small molecule ligand binding is characterized using a variety of methods, including surface plasmon resonance, isothermal titration calorimetry, and electrophoresis that all have different types of error and biases, making it difficult to easily compare binding information across aptamers. It is true, though, that knowledge of how or where the ligand binds to the aptamer can inform design choices. As our knowledge and ability to model ligand-aptamer interactions improve, this type of information can be incorporated into future iterations of this design tool. We have updated the Discussion section with this information.

3. The authors perform a comprehensive comparison of the performance of a lasso model with CNN and show that the CNN model consistently performs better, even on ribozymes with novel loop structures. However, use of the CNN and the sole comparisons with the lasso model are not well justified. The authors should include comparisons with other methods such as SVM, random forest, or perhaps even ensemble methods.

We thank the reviewer for this comment. Following the reviewer’s suggestion, we have also used these other methods to predict basal gene-regulatory activity from sequence on these same test and training sets. The Random Forest method performed well on the N4N4 library, producing predictions with an R^2^ value of 0.92, but much less well on the theophylline ribozyme library, with an R^2^ value of 0.06. The SVM method performed even worse, producing R^2^ values of 0.45 and 0.002, respectively. This further emphasizes the benefit of using a CNN model that can take into account the spatial relationship of the features when making predictions. We have included the performance of these models in the manuscript and as a supplementary figure (Supplementary Figure 4).

4. Have the authors considered the following explanation for why their method has limited success: If the aim is to increase the activation ratio, the high-throughput experiments should include the experiments in the presence of ligands. Seeking sequences of low gene-regulatory activities and hoping for high activation ratios will not work as this paper shows.

We thank the reviewer for this comment. For this work, the training dataset was constructed first, without containing any aptamer sequences to create a generalized set of ribozymes. Therefore, when we were designing the ribozyme switches, we did not have access to data on sequences with these aptamers until after we designed and constructed the switch libraries. While one could train a machine learning model on high-throughput screens in the presence and absence of ligands, the benefit of our work is that this type of high-throughput experimentation is not needed. Our model was developed such that a generalized dataset could act as training data for novel aptamers, without those aptamers having to be measured themselves. Our work demonstrates that the right model, when trained on this generalized dataset, can be used to design switches with activation ratios at levels useful for engineering purposes with reasonable success rates.

5. The authors use a wide range of loop lengths and features to construct libraries for training and initial testing of their model, but find that their algorithm does not work as well for complex libraries. Did the authors try training their model on complex libraries including the libraries of 174,080 ribozyme sequences paired with aptamers? If not, this might be worth trying out.

We thank the reviewer for this comment. The library of 174,080 ribozyme sequences paired with aptamers you are referring to were the sequences that were considered as potential switches. They were only screened in silico, and were never synthesized or had their gene-regulatory activity measured. We have altered the text to clarify that we do not have measurements for these sequences.

For the objectives of this study, it would not be proper to use these as a training set for the design of ribozyme switches with the same aptamers. This study was simulating the design of ribozyme switches with aptamers for which no ribozyme switches had been generated. Having a significant number of ribozymes in the training set sharing an aptamer with those in the design set would not give an accurate indication of the model’s ability to design novel ribozyme switches. However, now that libraries of ribozyme switches have been generated with each of these aptamers (Figure 5), they could be included in the training set for the future design of ribozymes with other aptamers and may increase the accuracy by further representing the influence of complex loop structures. We have updated the Discussion section with this point.

Other points:

1. The introduction section and the significance value of the manuscript is primarily focused on RNA switches. Although the goal of the study was to facilitate finding ribozyme switches, the CNN was not very successful at doing so, beyond a few switches. The authors should revisit their narrative to focus on the background and applicability of the major outcome of their tool: finding regulatory ribozymes.

We thank the reviewer for this comment. The goal of this study was to both demonstrate how machine learning can be applied to RNA sequence to predict regulatory activity and to provide a tool that allows for researchers to design their own regulatory ribozymes. Our paper demonstrated the first point by accurately predicting which ribozymes have low basal gene-regulatory activity, while demonstrating the second point by finding switches with statistically-significant changes in activity for 4 of the 5 aptamers tested as well as releasing that tool to the public through open-source software. Having created switches for 80% of the aptamers, we have shown that this is a forward design tool with value. We have changed the introduction section to emphasize this point.

2. While the application of CNN for ribozyme design is very interesting, the "back-box" nature of the models precludes a direct application of these methods for ribozyme engineering and design. From the results, it is unclear what specific sequence and conformational features are contributing to the observed accuracy in predicting ribozyme activity. The authors should expand on the "explainable" aspects of their model to substantiate their design/engineering claims.

We thank the reviewer for this comment. While it is preferable to have models that are more explainable, we used the CNN model because it had by far the best performance of comparable models. The fact that this type of model is less explainable does not preclude it from being used in forward design, as we show by using it to design myriad ribozyme switches.

We expanded on the model’s explainable aspects by examining the model’s preferences in terms of common sequence motifs for specific aptamers and nucleotide preference. The area of machine learning concerned with interrogating and explaining neural networks, especially neural networks as applied to genomics, is still new, but we could begin applying lessons from that field as it develops. This would not only begin to explain what contributes to predicted ribozyme activity, but because our model appears to be an accurate predictor of said activity, also provide further insights into the biophysical mechanisms governing ribozyme activity. We have updated the model exploration and conclusion sections with this information.

3. In the introduction, the authors refer to previously developed software for identifying RNA switches. It is unclear how the machine learning approach described here is better at predicting RNA switches than previously developed software tools pointed out by the authors. In fact, the other softwares identify more switches. The authors should bring out the benefits of using their algorithm over others.

We thank the reviewer for this comment. While other software tools also exist for designing RNA switches, ours is the first that works using the ribozyme platform. This is important not only because this platform is usable across eukaryotic systems, but also because, to advance our ability to design biological systems, having orthogonal systems of gene regulation allows for larger constructs to be built without genetic overlap, thus avoiding recombination-based dropout. In addition, the other design tools do not use machine learning, but rather biophysical models and rules for their design process. One of the benefits of a machine learning-based model is that it will only improve as the quantity and quality of the training data is improved. We have updated the manuscript to reflect these points more clearly.

4. By saying "However, none of the computational design tools described to date are based on gene regulatory platforms that function across a broad range of organisms.", it is implied that the current manuscript is also focused on finding broad-range switches. However, the new aptazymes described have been tested only in Yeast. It will make the manuscript better if the authors could test some of the high performing switches in human or other model organisms.

We thank the reviewer for this comment. While the ribozyme switches generated in this study were tested only in yeast, the hammerhead ribozyme platform has been proven to produce ribozyme switches that function in mammalian cells, including human cell lines such as HEK293T. Indeed, even the method for ribozyme switch discovery – attaching an aptamer to one stem, randomizing the other loop, and screening for functional switches – has been recently shown to produce switches in mammalian cells, using a high throughput sequence-based screening assay instead of the computational screen that we show here. We have added references to the manuscript to support this point.

5. The abstract and the end paragraph of introduction mention different number of aptamer switches that were identified. 5 in the abstract and 4 in introduction and later sections.

We thank the reviewer for this comment. We have corrected the abstract to be consistent with the introduction in that switches were identified for 4 out of 5 aptamers.

6. At multiple places in the manuscript the authors attribute the lack of ligand dependent switch responsiveness to impermeability of the cells to specific ligands. There is however no direct evidence for this reasoning. In some cases, these statements also lack references, for example "…including the ability of tetracycline to more efficiently cross the cell membrane."

We thank the reviewer for this comment. The cited paper Massively parallel RNA device engineering in mammalian cells with RNA-Seq demonstrated that in mammalian cells, overexpression of a folate transporter was necessary to enable full induction levels. We have added citations that directly demonstrate that tetracycline can diffuse through lipid membranes, while chloramphenicol and folinic acid both rely on active transporters.

7. Several parts of the result section describe data without referring to the associated table/Figure.

We thank the reviewer for this comment. We have updated the manuscript to include specific citations to Figure 2B and Figure 4B in the areas of the manuscript that discuss those results.

8. "The method starts by using the RNA secondary structure prediction program RNAstructure (21) to determine the sequence of loops I and II of the ribozyme (Figure 3A)." What exactly is meant, as RNAstructure employs sequences to predict secondary structure?

We thank the reviewer for this comment. Since the structure of the loop is unknown before the computational folding, it is also unknown which specific nucleotides are included in the loop as opposed to included in the base stem or any stem leading off of the loop (of particular importance when aptamers are included, as their larger size tends to lead to additional structures leading off of the loop). Therefore, since the sequence of the loop is not known until the structure of the loop is known, a structure-prediction tool such as RNAstructure determines both the structure and the sequence of the loop. We have modified the section of the manuscript concerned with feature development to clarify this point.

9. "Next, the sequences of the two loops are analyzed to determine the nucleotides and their positions in the loops". Not clear. If the sequence is known, would their positions in the loops be known?

We thank the reviewer for this comment. While the sequence of the loops is known, their position in the loop is not known until the structure is determined due to the fact that different loop sequences can produce different loop structures. One such example is if the first two nucleotides in the loop after the stem can form a Watson-Crick pair, which would extend the stem further and therefore make the following two nucleotides the actual first ones in the loop. Another example of this are ribozymes with longer loop sequences that can form stems extending off of the loop, which functionally removes some of the nucleotides from the loop. We have updated the text at the structural analysis section to clarify this point.

10. How are stem-loop interactions used as input in the 3-D image? Where does the interaction information come from? Was the input mapping onto the known 3D structure of hammerhead ribozymes? What about other ribozymes, do the authors input their 3D structures as well? Can the method be applied to a ribozyme with unknown 3D structures?

We thank the reviewer for this comment. The information used to create the 3D image are the predicted secondary structure and the sequence of the ribozyme. The sequence is known, and the secondary structure is predicted from this sequence. It is then formed into a 3-dimensional matrix that nominally represents the position of the loop nucleotides in the tertiary structure. No explicit mapping onto a known or predicted tertiary structure is performed, however the crystal structure of a native hammerhead ribozyme was used to develop the 3D matrix. Since this design tool uses only the hammerhead ribozyme sTRSV as a base for engineering, we did not take into account other ribozymes with known or unknown 3D structures. This method would not be able to be applied to ribozymes other than the hammerhead ribozyme as their 3D structures are quite different, and therefore have different nucleotide interactions that lead to cleavage.

11. Is the average expression for 15 ribozymes shown in Figure 6A significantly different in the presence or absence of ligands? Box plots in Figure 6A needs p-values.

We thank the reviewer for this comment. We have added p-values to the box plots, but would like to emphasize that what is important is the change in activity between individual ribozyme sequences.

12. Since Figure 6B is a reanalysis of data shown in Fig6A, it can be moved to supplementary. Figure 6A will be more informative if the points were connected between the two box plots so the readers can observe switch in activity for every plotted ribozyme.

We thank the reviewer for this comment. Given that there are 15 points in each box plot, we believe that connecting them would produce a confusing plot. In addition, since claims are made about the importance of Activation Ratio values above 3, we believe that it is important to directly showcase the distribution obtained by our design tool. We have included a plot that connects the dots as a supplementary figure (Supplementary Figure 9).

13. The high R-squared values (0.91 and 0.82) in Figure 4B are a bit surprising considering the poor fit especially in the head and tail regions. The authors need to check their fit and R-squared calculations. Likewise, R-square values reported in supplementary Figure 1A (0.92) and 1B (R-squared = 0.94) needs to be cross checked for accuracy.

We thank the reviewer for this comment. Having rechecked our calculations, the fit numbers are indeed accurate, according to the Scipy.stats linregress function. The large number of sequences included in the test set (for Figure 4B) and found in the FACS-Seq experiments (for Supplementary Figure 1A) can make it look like there are a lot of outliers, even if it is still a small percentage.

[Editors' note: further revisions were suggested prior to acceptance, as described below.]

The manuscript has been improved but there is a smallish remaining issue that should be addressed:

Mean squared errors of the test set are still at least three times larger than those of the training set (Figure 3—figure supplement 1). Thus, the method is clearly biased toward the training set and it is not that clear how generalisable it is. It is a bit difficult to understand why this should be an advantage rather than a negative. In any case, please discuss and emphasize in the abstract and in the Discussion section, so that the reader has a clear sense of the limitation of the technique.

It is true that, just as in any complex model with a significant number of features, the training set has a lower error than the test set. In our paper, we explore the reasons behind this to show that a significant source of bias comes from differences in the structures of ribozymes in the test set vs. ribozymes in the training set. Our training set of ribozymes partially rectifies this by including ribozymes with a diverse set of structures, but potential sources of bias remain. We have modified the abstract and Discussion sections to further emphasize the model’s bias towards the training set and the subsequent impact on generalizability.